# Interpreting Molecule Generative Models for Interactive Molecule Discovery

## Abstract

Discovering novel molecules with desired properties is crucial for advancing drug discovery and chemical science. Recently deep generative models can propose new molecules by sampling random vectors from latent space and then decoding them to a molecule structure. However, through the feedforward generation pipeline, it is difficult to reveal the underlying connections between latent space and molecular properties as well as customize the output molecule with desired properties. In this work, we develop a simple yet effective method to interpret the latent space of the learned generative models with various molecular properties for more interactive molecule generation and discovery. This method, called **Molecular Space Explorer (MolSpacE)**, is model-agnostic and can work with **any** pre-trained molecule generative models in an off-the-shelf manner. It first identifies *latent directions* that govern certain molecular properties via the *property separation hyperplane*, and then moves molecules along the directions for smooth change of molecular structures and properties. This method achieves interactive molecule discovery through identifying interpretable and steerable concepts that emerge in the representations of generative models. Experiments show that MolSpacE can manipulate the output molecule toward desired properties with high success. We further quantify and compare the interpretability of multiple state-of-the-art molecule generative models. An interface and a demo video are developed to illustrate the promising application of interactive molecule discovery.

## 1 Introduction

Designing molecules with desired properties is a fundamental problem in chemistry, which has a variety of applications in drug discovery and material science (Chen et al., 2018). Traditional pipelines require exhaustive human efforts and domain knowledge, which are difficult to scale up. Recent studies exploit deep generative models to solve this problem by encoding molecules into a latent space, from which random samples are drawn and decoded to novel molecules (Walters & Barzilay, 2020). It has been widely observed that such deep molecule generative models are able to facilitate the design and development of drugs and materials from many perspectives (Lopez et al., 2020; Sanchez-Lengeling & Aspuru-Guzik, 2018).

Despite the promising results of deep generative models for molecule generation, much less effort has been made to interpret the learned representations. Most of the existing models are based on deep neural networks, which are known to be short on interpretability (Samek et al., 2019). Outside of molecule generation domain, many attempts have been made to improve the interpretability of deep learning models from various aspects, *e.g.*, representation space (Zhou et al., 2016), model space (Guo et al., 2021), and latent space (Shen et al., 2020; Shen & Zhou, 2021). In molecule generation domain, interpretability can be studied in two ways: (1) the interpretation of **learned latent space** where steering the value of latent vectors could lead to smooth and continuous molecular property change and (2) the interpretation of **molecular space** that adjusting the molecular property could observe smooth structure change of molecules.

In addition, it remains challenging to generate molecules with desired properties. Previous works mostly rely on optimization-based, reinforcement learning-based, and searching-based methods to achieve property control of the generated molecules (Shi et al., 2020; Jin et al., 2018a). Specifically, reinforcement learning-based algorithm (You et al., 2018a) equips the model with rewards designed

to encourage the molecule generative models to generate molecules with specific molecular properties. Optimization-based algorithm takes advantage of the learnt latent space by molecule generative models and optimize the molecular properties via Bayesian Optimization (Liu et al., 2018). Searching-based algorithm instead searches directly from the chemical space for molecules with optimal properties (Kwon et al., 2021). However, these lines of work are designed for molecule generation with optimized property and thus unable to change the property monotonically and smoothly. Besides, current methods are confined to a limited number of molecular properties, which hinders real-world applications in drug discovery and material science. For example, existing work only discover a limited set of molecular properties, such as penalized logP (octanol-water partition coefficient), QED (Drug-likeness), DRD2 activity, etc (Jin et al., 2018a; Shi et al., 2020; Liu et al., 2018; Fu et al., 2020). Consequently, when molecules with new properties are needed, the models must be re-trained with a different optimization goal, which is significantly time-consuming.

To tackle the above challenges, we formulate a new task, *molecule manipulation*, which aims to improve the interpretability and steerability of a given molecule generative model via continuously manipulating molecular properties. Based on the observation that molecules sharing similar structures/properties tend to cluster in the latent space, we develop *MolSpace Explorer*, a model-agnostic method to manipulate molecules with continuous changes of molecular properties. Specifically, *MolSpace Explorer* first identifies the *property separation hyperplane* which defines the boundary for molecular properties (*e.g.*, drug-like or drug-unlike) in the latent molecular space learned by a given generative model. Based on the property separation hyperplane, we estimate the *latent directions* that govern molecular properties, which are in turn used to enable continuous change of the molecular structures and properties without re-training the given molecular generative model. To the best our knowledge, this work is one of the earliest attempts to achieve interactive molecule discovery through the steering of pretrained generative models.

The experiments demonstrate that our method can effectively quantify the interpretability and steerability of state-of-the-art molecule generative models. To measure the ability of generative models in interpreting molecular properties and generating molecules with continuous property control, we design a new evaluation metric named *success rate*, which evaluates the percentage of successful manipulations with continuous property-changing molecules over manipulations of a group of molecules. To visualize our method and facilitate interactive molecule discovery for scientists, we develop an interactive system with visualization of real-time molecule manipulations and smooth molecular structure/property changes. Our main contributions are summarized as follows:

- We formulate *molecule manipulation*, a new task which measures the interpretability and steerability of molecule generative models via the ability to manipulate the molecular properties of molecules in the latent space.

- We develop a simple yet effective model-agnostic method named *MolSpace Explorer* for molecule manipulation, which further analyzes current molecule generative models in terms of their interpretability and steerability.terpretation.

- Comprehensive experiments demonstrate the effectiveness of our method in quantifying the interpretability and steerability of various molecule generative models. An interactive system is developed for real-time molecule manipulation.

## 2 RELATED WORK

**Molecule Generation.** Recent studies have explored a variety of deep generative models for molecule generation. Specifically, GrammarVAE (Kusner et al., 2017) designs a variational autoencoder-based model that represents molecules as SMILE strings. With the advancement of graph neural networks (GNN), a surge of GNN-based generative models have been proposed to tackle the problem, by combining GNN with variational autoencoders (VAEs), generative adversarial networks (GANs), normalizing flows, energy-based models (EBMs), and reinforcement learning (Olivecrona et al., 2017; De Cao & Kipf, 2018; Jin et al., 2018a; Zhou et al., 2019; Madhawa et al., 2019; Shi et al., 2020; Luo et al., 2021; Liu et al., 2021; Yang et al., 2021). To be specific, JT-VAE (Jin et al., 2018a) proposes a VAE-based architecture to encode both atomic graphs and structural graphs for efficient molecule generation. MolGAN (De Cao & Kipf, 2018) exploits GANs for molecule generation, where discriminators are used to encourage the model to generate realistic and chemically-valid molecules. MRNN (Popova et al., 2019) extends the idea of GraphRNN (You

et al., 2018b) to formulate molecule generation as an auto-regressive process. GCPN (You et al., 2018a) formulates the molecule generation process as a reinforcement learning problem where it obtains a molecule step by step by connecting atoms and reward is used for controllable generation. GraphNVP (Madhawa et al., 2019) first introduces normalizing flows for molecule generation, where the generation process is invertible. Later work improve the flow-based models via auto-regressive generation (Shi et al., 2020), valency correction (Zang & Wang, 2020), and discrete latent representation (Luo et al., 2021). GraphEBM (Liu et al., 2021) introduces energy-based models which models the density of molecule data.

**Controllable Molecule Generation.** Another key point for molecule generation is controllable generation where the generated molecules are expected to possess certain properties. Early work (Segler et al., 2018) bias on the distribution of the data and fine-tune the generative models with known desired properties to generate molecules with desired properties. The recent work mainly leverage optimization-based (Shi et al., 2020; You et al., 2018a; Hoffman et al., 2020; Winter et al., 2019;?), reinforcement learning-based (Zang & Wang, 2020; Jin et al., 2018a; Blaschke et al., 2020), and searching-based (Brown et al., 2019; Yang et al., 2020; Kwon et al., 2021) approaches to generate molecules with desired properties. Optimization-based methods are quite flexible and can both directly work on the molecules (Renz et al., 2019; Fu et al., 2020; Xie et al., 2021; Maziarz et al., 2021) and work on the learnt latent vectors of the molecules (Gómez-Bombarelli et al., 2018; Jin et al., 2018b; Winter et al., 2019; Griffiths & Hernández-Lobato, 2020; Notin et al., 2021). Reinforcement learning-based methods usually formulates controllable generation as a sequential decision-making problem and requires a score-function to provide rewards to the agent. Searching-based approaches (Brown et al., 2019; Yang et al., 2020; Kwon et al., 2021) are also capable of searching molecules with optimized properties. Besides, few work (Chenthamarakshan et al., 2020; Das et al., 2021) leverage the learnt latent space and achieve controllable generation by accepting/rejecting sampled molecules based on a molecular property predictor. Despite the ability to generate molecules with optimized properties, it is challenging for existing methods to interpret the generation process and cannot generate molecules with monotonically and smoothly changing molecular properties.

## 3 PRELIMINARIES

**Molecule Graph.** Molecules can be presented as graphs $X = (\mathcal{V}, \mathcal{E}, E, F)$, where $V$ denotes a set of $N$ vertices (*i.e.*, atoms), $\mathcal{E} \subseteq V \times V$ denotes a set of edges (*i.e.*, bonds), $F \in \{0,1\}^{N \times D}$ denotes the node features (*i.e.*, atom types) and $E \in \{0,1\}^{N \times N \times K}$ denotes the edge features (*i.e.*, bond types). The number of atom types and bond types are denoted by $D$ and $K$, respectively.

**Deep Molecule Generative Models.** In molecule generation, a generative model $M$ encodes the molecular graph $X$ as a latent vector $Z \in \mathbb{R}^l$ with $l$ being the dimension of the latent space, and is capable of decoding any latent vector back to the molecular space. Specifically, variational auto-encoder (VAE) (Kingma & Welling, 2013) and flow-based model (Flow) (Rezende & Mohamed, 2015) are the two most commonly used models for molecule generation tasks. Both of them encode the data from molecular space to latent space, which is usually modeled as a Gaussian distribution; then they decode the latent code back to molecular space. They can be formulated as:

$$z = f(x), \qquad x' = g(z), \qquad (1)$$

where $x$ and $x'$ are the ground-truth and reconstructed/sampled data respectively, and $z \in Z$ represents a latent vector in the latent space.

## 4 PROBLEM FORMULATION OF MOLECULE MANIPULATION

To improve the steerability and interpretability of molecule generative models, we propose a new research task, *molecule manipulation*, which interprets the generative model and steers the properties of the output molecules. To be specific, a deep generative model contains a generator $g: \mathcal{Z} \to \mathcal{X}$, where $\mathcal{Z} \in \mathbf{R}^l$ stands for the $l$-dimensional latent space, which is commonly assumed to be Gaussian distribution (Kingma & Welling, 2013; Rezende & Mohamed, 2015). There exist property functions $f_P$ which define the property space $\mathcal{P}$ via $P = f_P(X)$.

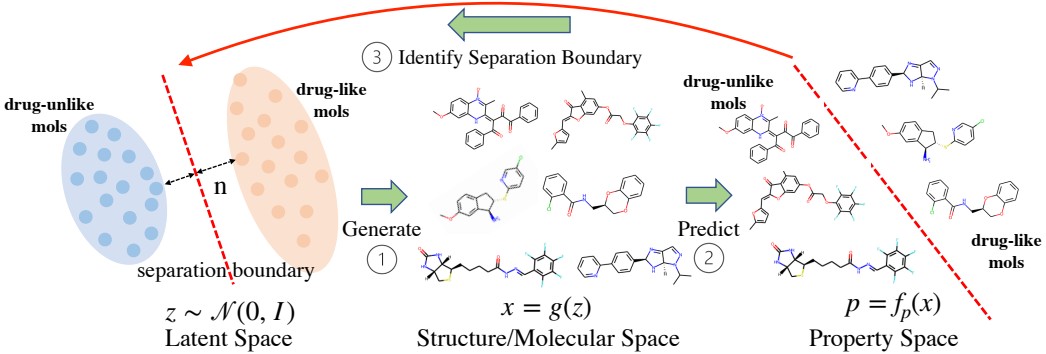

Figure 1: *MolSpacE* framework: (1) the tested molecule generative model generates novel molecules by sampling random vector from the latent space and then feeding it into the generator, (2) off-the-shelf property prediction function is used to predict molecular properties from the molecular space, (3) MolSpacE identifies latent directions which govern molecular properties via the property separation hyperplane.

**Formulation.** The input to molecule manipulation is a list of $n$ molecules $X = \{x_1, x_2, \cdots, x_n\}$ and a list of $m$ molecular properties $P = \{p_1, p_2, \cdots, p_m\}$. We aim to manipulate one or more molecular properties $p$ of a given molecule in a $k$ consecutive steps and output the manipulated molecules with properties $p' = \{p^{(1)}, p^{(2)}, \cdots, p^{(k)}\}$. By manipulating the given molecule, we can observe the alignment of $\mathcal{Z} \rightarrow \mathcal{X} \rightarrow \mathcal{P}$, where the relationship between $\mathcal{Z}$ and $\mathcal{X}$ explains the latent space of molecule generative models. The relationship between $\mathcal{X}$ and $\mathcal{P}$ reveals the correlations between molecular structures and properties. By traversing latent space, we can generate molecules with continuous structure/property changes.

**Evaluation.** For the molecule manipulation task, we design two new evaluation metrics named *success rate (SR)* and *soft success rate (SSR)* that measure the performance in discovering latent molecular property directions. To be specific, we consider a manipulation to be successful only if we generate molecules with monotonically-changing properties in a consecutive $k$ steps of manipulation, as follows:

$$\phi_{success}(x, k) = \mathbb{1}[\forall\, i \in [k], s.t., f_p(x^{(i)}) - f_p(x^{(i+1)}) \leq 0], \tag{2}$$

where $f_p$ is a property function which calculates certain molecular property and $x^{(i)}$, $x^{(i+1)}$ represents molecules generated in two adjacent steps. As monotonicity is rather strict, we propose a more flexible definition of success, namely soft success, as follows:

$$\phi_{soft\ success}(x, k) = \mathbb{1}[\forall\, i \in [k], s.t., f_p(x^{(i)}) - f_p(x^{(i+1)}) \leq \epsilon], \tag{3}$$

where $\epsilon$ is a predefined tolerance threshold that weakens the monotonicity requirement. To extend the evaluation metric for $|P|$ molecular properties and $|X|$ candidate molecules to manipulate, we calculate the overall SR as:

$$SR(P, X, k) = \frac{\sum_{p \in P, x \in X} \mathbb{1}[\phi_{success}(x_p, k) \wedge \phi_{SD}(x_p, k) \wedge \phi_{DIV}(x_p, k)]}{|P| \times |X|}, \tag{4}$$

where $x_p$ represents manipulating property $p$ of molecule $x$ which results in a manipulation path $x_p = \{x_p^{(i)} | i \in [k]\}$. Since the molecular space is essentially discrete, we allow the model to generate duplicate molecules during manipulation, but the model has to generate at least one distinct molecule from the base molecule (diversity or DIV) and the structure difference (SD) enforces monotonically-decreasing structure similarity along the manipulation sequence, as follows:

$$\phi_{DIV}(x, k) = \mathbb{1}[\exists\, i \in [k], s.t., x^{(i)} \neq x^{(1)}], \tag{5}$$

$$\phi_{SD}(x, k) = \mathbb{1}[\forall\, i \in [k], s.t., \delta(x^{(i)}, x^{(1)}) - \delta(x^{(i+1)}, x^{(1)}) \geq 0], \tag{6}$$

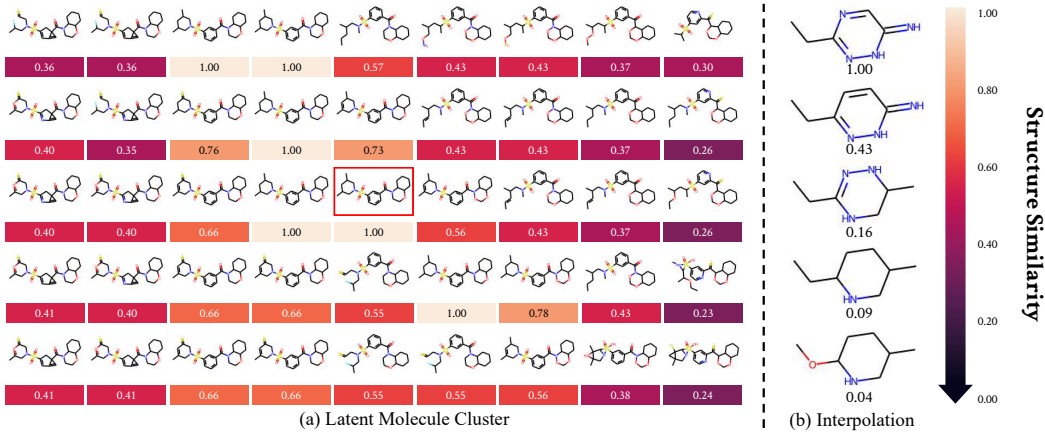

(a) Latent Molecule Cluster    (b) Interpolation

Figure 2: (a) Molecule clusters in the latent space, the number represents structure similarity (Bajusz et al., 2015), where the red box represents the base molecule, x and y axes denote two random orthogonal directions to manipulate. (b) Linear interpolation of two (top and bottom) molecules.

where $\delta$ measures the structure similarity between the two input molecules. Similarly, we provide a soft structure difference (SSD) with a predefined tolerance threshold $\gamma$ as follows:

$$\phi_{SSD}(x, k) = \mathbb{1}[\forall\, i \in [k], s.t., \delta(x^{(i)}, x^{(1)}) - \delta(x^{(i+1)}, x^{(1)}) \geq \gamma], \tag{7}$$

Finally, we calculate the overall SSR as:

$$SSR(P, X, k) = \frac{\sum_{p \in P, x \in X} \mathbb{1}[\phi_{soft\ success}(x_p, k) \wedge \phi_{SSD}(x_p, k) \wedge \phi_{DIV}(x_p, k)]}{|P| \times |X|}, \tag{8}$$

'

## 5 MOLECULE MANIPULATION THROUGH MOLSPACE

### 5.1 LATENT CLUSTER ASSUMPTION

We examine the property of latent space learned by the generative models and have the following observations, (1) molecules with similar structures tend to cluster in the latent space, (2) interpolating two molecules $x_1$ and $x_2$, represented by latent vectors $z_1$ and $z_2$, can lead to a list of intermediate molecules whose structures gradually change from $x_1$ to $x_2$. As molecular structures determine molecular properties (Seybold et al., 1987), the observations imply that molecules with similar molecular properties cluster together and interpolating two molecules could lead to gradual changes in structure. As shown in Fig. 1, there may exist two groups of molecules, drug-like and drug-unlike, where each group cluster together and linear interpolating two latent vectors with one molecule from each group could lead to a direction that crosses the separation boundary. These observations also match the analysis from the prior work (Zang & Wang, 2020). To verify our assumption, we visualize the latent space of the pre-trained MoFlow model trained on QM9 dataset in Fig. 2. The left figure shows that molecules that are close in the latent space are similar in structures. The right figure shows that interpolating two molecules in the latent space could lead to smooth structure changes.

### 5.2 IDENTIFYING LATENT DIRECTIONS

**Latent Separation Boundary.** With the verifications above and the previous work of analyzing the latent space of generative models (Shen et al., 2020; Bau et al., 2017; Jahanian et al., 2019; Plumerault et al., 2020), we assume that there exists a separation boundary which separates groups of molecules (*e.g.*, drug-like and drug-unlike) and the normal vector of the separation boundary defines a latent direction (in Fig. 1). When $z$ moves toward and crossing the boundary, the molecular

properties change accordingly (*e.g.*, from drug-unlike to drug-like). A perfect separation boundary would have molecules with different properties well separated on different sides. From that, we can find a hyperplane for each molecular property with a unit normal vector $n \in \mathbf{R}^l$, such that the distance from any sample $z$ to the hyperplane as:

$$d(z, n) = n^T z \tag{9}$$

**Latent Direction.** In the latent space, the molecular structure and property change continuously towards the new property class when $z$ moves towards the hyperplane and vice versa, where we assume linear dependency between $z$ and $p$:

$$f_P(g(z)) = \alpha \cdot d(z, n), \tag{10}$$

where $f_P$ is a property function and $\alpha$ is a degree scalar that scales the changes along that corresponding direction. Extending the method to multiple molecular properties manipulation, we have:

$$f_P(g(z)) = AN^T z, \tag{11}$$

where $A = Diag(a_1, \cdots, a_m)$ is the diagonal matrix with linear coefficients for each of the $m$ molecular properties and $N = [n_1, \cdots, n_m]$ represents normal vectors for the separation boundaries of $m$ molecular properties. We have the molecular properties $P$ following a multivariate normal distribution via:

$$\mu_P = \mathbf{E}(AN^T z) = AN^T \mathbf{E}(z) = \mathbf{0}, \tag{12}$$

$$\Sigma_P = \mathbf{E}(AN^T zz^T NA^T) = AN^T \mathbf{E}(zz^T)NA^T = AN^T NA^T. \tag{13}$$

We have all disentangled molecular properties in $P$ if and only if $\Sigma_P$ is a diagonal matrix and all directions in $N$ are orthogonal with each other. Nevertheless, not all molecular properties are purely disentangled with each other. In that case, molecular properties can correlate with each other and $n_i^T n_j$ is used to denote the entanglement between the $i$-th and $j$-th molecular properties in $P$.

## 5.3 IMPLEMENTING THE MOLECULE MANIPULATION

After we find the separation boundary and identify the latent direction, to manipulate the generated molecules with desired properties, we first move from latent vector $z$ along the direction $n$ with a degree scalar $\alpha$, and the new latent vector is

$$z' = z + \alpha n \tag{14}$$

To this end, the expected property of the new manipulated molecule is

$$f_P(g(z + \alpha n)) = f_P(g(z)) + k\alpha \tag{15}$$

where $k$ is a scaling factor between molecular vector space and property. Based on our assumption to find a separation boundary for each molecular property, we utilize linear Support Vector Machine (SVM) (Cortes & Vapnik, 1995) to find the separation boundaries which best separate the two classes of the data. For each molecular property, we train an individual linear SVM from a group of randomly sampled latent vectors and utilize a property function $f_P$ to calculate the corresponding molecular properties. Then, we find the separation boundaries for each molecular property. The normal vectors $N$ of separation boundaries are finally utilized as identified latent directions that govern the molecular properties. Additionally, our method is highly efficient thanks to leveraging shallow models compared to optimization-based methods which require re-training of deep models for each of the specific molecular properties.

## 6 EXPERIMENTS

## 6.1 EXPERIMENT SETUP

**Datasets.** We use three molecule datasets, QM9 (Ramakrishnan et al., 2014), ZINC250K (Irwin & Shoichet, 2005), and ChEMBL (Mendez et al., 2019). QM9 contains 134k small organic molecules with up to 9 heavy atoms (C, O, N, F). ZINC is a free database of commercially-available compounds for drug discovery. On average, the molecules in ZINC are bigger ($\sim$23 heavy atoms)

and structurally more complex than QM9. We take a sampled 250K molecules version (Gómez-Bombarelli et al., 2018) from the larger database. ChEMBL is a manually curated database of bioactive molecules with drug-like properties and contains ∼1.8 million molecules.

**Baselines.** We include two baseline methods of identifying latent direction that governs the molecular property for comparisons. **Random manipulation** randomly samples latent directions for molecular properties. **Largest range manipulation** draws latent vectors from the training set and defines the directions via calculating the direction between one molecule with the largest property score and another molecule with the smallest property score for each molecular property.

**Implementation Details.** We take the publicly available pre-trained models from the GitHub Repository of HierVAE, and MoFlow, respectively. For CGVAE, we take the code implementation from the public GitHub Repository and train using default hyper-parameters provided by the authors. We utilize the SVM implementation from Scikit-Learn[1].

## 6.2 EVALUATION PROTOCOLS

**Pre-trained Models.** We apply MolSpacE, as well as baselines, on three state-of-the-art molecule generative models with publicly available pre-trained models or code implementations. Specifically, CGVAE (Liu et al., 2018) implements a VAE-based model and generate molecular graphs in a sequential manner; HierVAE (Jin et al., 2020) embeds molecular structure motifs into a hierarchical VAE-based generative model; MoFlow (Zang & Wang, 2020) designs a normalizing flow-based model which learns an invertible mapping between input molecules and latent vectors.

**Molecular Properties.** We study molecular properties identified in the chemistry community through open-source cheminformatics software[2]. In total, we analyze 154 molecular properties from multiple dimensions, including distributions, inter-correlations, etc. Details can be found in Appendix B. Due to the page limit, we mainly report results for 7 molecular properties, including 3 very common yet important ones, drug-likeness (QED), molecular weight (MolWt), partition coefficient (LogP), and 4 randomly selected ones, including topological indexes, BalabanJ, BertzCT, and BCUT2D descriptors (Bayada et al., 1999) CHGHI, CHGLO.

Quantitatively, we evaluate the ability of the model to manipulate the given molecular property of molecules with the proposed **strict success rate** and **success rate** metrics (see Sec. 4). Qualitatively, we visualize molecule manipulation including property distribution shift during manipulation, single and multiple property manipulations.

## 6.3 QUANTITATIVE EVALUATION OF MOLECULE MANIPULATION

In Table 1, we report the quantitative evaluation results for molecule manipulation with both strict success rate and success rate, which are evaluated on 154 molecular properties. According to the table, we can obtain the following insights.

(1) Our proposed method, MolSpacE, as the first attempt for molecule manipulation, achieves excellent performance to manipulate properties of molecules with three state-of-the-art molecule generative models. For some important molecular properties (*e.g.*, QED), we (with HierVAE) achieve 68% manipulation success rate in QM9 dataset. We outperform the baseline methods 80% on average.

(2) Generally, HierVAE performs the best in all three datasets. We conjecture this is because HierVAE models not only atomic level but also structural level information, which is critical in determining molecular properties. MoFlow ranks second due to the invertible constraint which maps molecular space to a latent space that reflects the property space. CGVAE underperforms on three datasets. One possible explanation is its simple architecture which struggles to map molecular space to a semantically meaningful latent space, where the molecular property is not well-reflected.

(3) CGVAE has the largest gap between SSR and SR, which demonstrates that CGVAE tends to repeatedly sample molecules with the same properties during manipulation, and this is possible when the latent space does not well separate molecules via properties. The SSR and SR gaps in MoFlow and HierVAE are similar and ∼ 25% smaller than QM9. For individual molecular properties, we

---

[1] https://scikit-learn.org/
[2] https://www.rdkit.org/docs/index.html

Table 1: Quantitative Evaluation of Molecule Manipulation over a variety of molecular properties (numbers reported are *strict success rate / success rate* in %, -R denotes model with random manipulation, -L denotes model with largest range manipulation, -M denotes model with MolSpacE. The best performances compared to baseline models are bold and the best performances across models in the same dataset are marked as red).

| Datasets | Models | Avg. | QED | MW | LogP | BalabanJ | BertzCT | CHGHI | CHGLO |
|---|---|---|---|---|---|---|---|---|---|
| QM9 | CGVAE-R | 1.75 / 16.64 | 0.70 / 0.70 | 1.50 / 1.50 | 0.90 / 1.10 | 1.60 / 1.60 | 1.50 / 1.50 | 0.70 / 0.70 | 1.10 / 1.10 |
| | CGVAE-L | 1.14 / 8.21 | 0.00 / 0.00 | 0.40 / 0.40 | 0.00 / 0.00 | 0.20 / 0.20 | 0.00 / 0.00 | 0.20 / 0.20 | 0.10 / 0.10 |
| | CGVAE-C | / | / | / | / | / | / | / | / |
| | CGVAE-M | **7.98** / **57.59** | **11.80** / **12.00** | **11.20** / **12.10** | **11.10** / **11.30** | **16.30** / **18.40** | **11.40** / **11.80** | **14.10** / **14.10** | **10.50** / **10.50** |
| | MoFlow-R | 10.40 / **47.97** | 4.40 / 4.40 | 13.60 / 13.60 | 9.30 / 9.30 | 20.30 / 22.80 | 7.30 / 7.30 | 9.60 / 9.60 | 9.10 / 9.10 |
| | MoFlow-L | 0.35 / 0.41 | 1.20 / 1.20 | 0.00 / 0.00 | 4.10 / 4.10 | 0.00 / 0.00 | 0.00 / 0.00 | 0.00 / 0.00 | 0.00 / 0.00 |
| | MoFlow-M | **16.62** / 41.38 | **39.90** / **39.90** | **44.10** / **44.20** | **40.40** / **40.40** | **21.00** / **27.50** | **30.90** / **32.20** | **29.10** / **29.10** | **37.00** / **37.00** |
| | HierVAE-R | 0.19 / 1.16 | 0.10 / 0.10 | 0.00 / 0.00 | 0.10 / 0.10 | 0.30 / 0.30 | 0.30 / 0.30 | 0.40 / 0.40 | 0.20 / 0.20 |
| | HierVAE-L | 0.24 / 0.87 | 0.10 / 0.10 | 0.20 / 0.20 | 0.00 / 0.00 | 0.10 / 0.10 | 0.30 / 0.30 | 0.00 / 0.00 | 0.10 / 0.10 |
| | HierVAE-M | *27.30* / *68.84* | *50.20* / *50.70* | *48.50* / *51.20* | *53.80* / *54.70* | *49.60* / *50.60* | *53.40* / *54.80* | *55.60* / *55.60* | *53.50* / *53.50* |
| ZINC | CGVAE-R | 0.72 / 6.18 | 0.10 / 0.10 | 0.10 / 0.10 | 0.10 / 0.10 | 0.00 / 0.00 | 0.20 / 0.20 | 0.10 / 0.10 | **0.30** / **0.30** |
| | CGVAE-L | 0.43 / 2.94 | 0.00 / 0.00 | 0.00 / 0.00 | 0.00 / 0.00 | 0.00 / 0.00 | 0.00 / 0.00 | 0.10 / 0.10 | 0.10 / 0.10 |
| | CGVAE-C | / | / | / | / | / | / | / | / |
| | CGVAE-M | **3.47** / *46.91* | **1.00** / **1.00** | **0.40** / **0.40** | **0.70** / **0.70** | **1.00** / **1.00** | **0.80** / **0.80** | **0.40** / **0.40** | 0.20 / 0.20 |
| | MoFlow-R | 2.96 / 18.00 | 0.90 / 0.90 | 0.80 / 0.80 | 1.50 / 1.50 | 1.90 / 2.10 | 2.00 / 2.00 | 0.60 / 0.60 | 0.60 / 0.60 |
| | MoFlow-L | 0.03 / 0.11 | 0.00 / 0.00 | 0.00 / 0.00 | 0.00 / 0.00 | 0.00 / 0.00 | 0.00 / 0.00 | 0.00 / 0.00 | 0.00 / 0.00 |
| | MoFlow-M | **9.61** / **24.43** | **21.80** / **21.80** | **25.00** / **25.10** | **28.80** / **29.00** | **16.80** / **21.10** | **21.70** / **22.00** | **19.30** / **19.30** | **19.20** / **19.20** |
| | HierVAE-R | 0.45 / 1.26 | 0.50 / 0.50 | 0.80 / 0.80 | 0.90 / 0.90 | 0.80 / 0.80 | 1.20 / 1.20 | 0.40 / 0.40 | 1.00 / 1.00 |
| | HierVAE-L | 0.70 / 0.91 | 0.90 / 0.90 | 0.40 / 0.50 | 0.50 / 0.50 | 1.10 / 1.10 | 1.00 / 1.00 | 0.70 / 0.70 | 0.70 / 0.70 |
| | HierVAE-M | *19.40* / *42.67* | *30.80* / *31.90* | *32.60* / *33.90* | *27.00* / *27.50* | *35.00* / *35.90* | *27.80* / *28.10* | *35.30* / *35.30* | *32.20* / *32.20* |
| ChEMBL | HierVAE-R | 0.13 / 0.93 | 0.00 / 0.00 | 0.10 / 0.10 | 0.10 / 0.10 | 0.10 / 0.10 | 0.20 / 0.20 | 0.10 / 0.10 | 0.00 / 0.00 |
| | HierVAE-L | 0.36 / 1.02 | 0.20 / 0.20 | 0.50 / 0.50 | 0.10 / 0.10 | 0.00 / 0.00 | 0.30 / 0.30 | 0.00 / 0.00 | 0.00 / 0.00 |
| | HierVAE-M | *19.59* / *62.38* | *17.40* / *17.50* | *28.30* / *28.40* | *22.50* / *22.80* | *15.40* / *15.60* | *25.00* / *25.20* | *23.70* / *23.70* | *24.80* / *24.80* |

can observe that the gaps between SSR and SR are small, while the average gap is large, we find that the gaps are large in categorical properties shown in Appendix D.3 Table 7 and 8.

(4) The baseline (random manipulation) method sometimes "finds" directions that control molecular properties. As shown in Fig. 2, the molecules are well-clustered in the latent space with respect to structures that determine molecular properties (Seybold et al., 1987). However, the largest range manipulation works worse possibly due to its strong assumption in determining the direction via the two extreme (largest property and smallest property) molecules in the dataset.

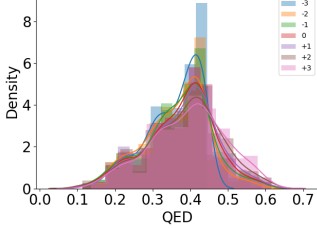 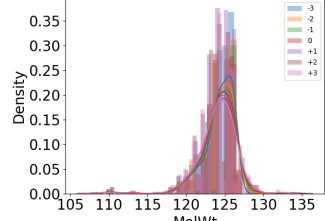 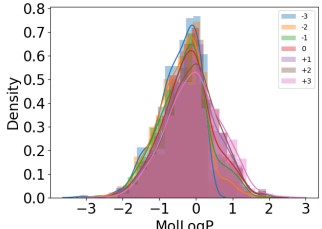

Figure 3: Visualization of Molecular property distribution shift while manipulating molecules with MoFlow on QM9 dataset (0 denotes the randomly sampled base molecule and $+x$ and $-x$ denote manipulation directions and steps).

## 6.4 QUALITATIVE EVALUATION OF MOLECULE MANIPULATION AND INTERPRETATION

In Fig. 11, we visualize the property distributions of QED, MolWt and LogP along a 7-step manipulation path. For each step, we draw a property distribution. The candidate molecules are at place 0 and we attempt to manipulate the molecular property to the left (lower) and the right (higher). From the figure, we can clearly observe that the property distribution shifts to the left and right accordingly when we manipulate the molecule to the left and right. For example, when we manipulate the molecules three steps to the left, the range of QED shifts from $[0, 0.7]$ to $[0, 0.5]$; when the molecules are manipulated three steps to the right, there are much more molecules that have QED > 0.5 than the base distribution. Similar trends can also be seen for MolWt and LogP properties.

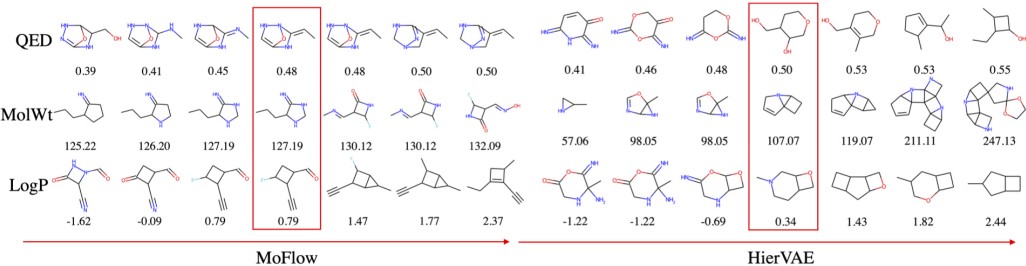

Figure 4: Manipulating QED, MolWt and LogP properties of sampled molecules. The backbone model is CGVAE trained on QM9 dataset.

**Single Property Manipulation.** To qualitatively evaluate the performance of our method for molecule manipulation, we randomly select the successful manipulation paths from all three generative models in Fig. 4. The figures have shown that our method succeeds in learning interpretable and steerable directions. For example, for HierVAE in Fig. 4, we can find that gradually increasing LogP of a molecule may lead to the removal of the heavy atoms $O$ and $N$ from the structure. With respect to QED, the molecule drops double bonds, as well as heavy $N$ and $O$ atoms, when increasing QED for the HierVAE model. A similar trend can be observed in the MoFlow model that increasing QED drops double bonds and $O$ atoms on the left of Fig. 4.

**Multi-Property Manipulation.** When it comes to multi-property manipulation, the goal is to control multiple molecular properties of a given molecule at the same

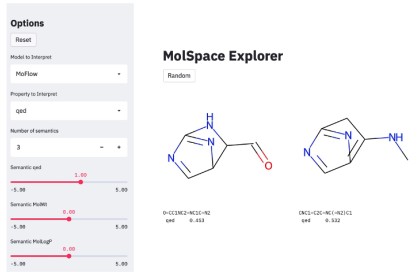

Figure 5: Manipulating QED and LogP properties of sampled molecules simultaneously with MoFlow model trained on QM9 dataset (the repeated molecules are removed for better visualization).

time. In Fig. 5, we show how our method can manipulate multiple molecular properties. For simplicity, we remove the duplicate molecules and only leave the different molecules during the manipulation. From the figure, we can observe that there are some correlations between LogP and QED since when we increase QED, LogP also increases accordingly. However, it is not always the case as moving the molecules to the right in the second row does not increase the QED scores. One potential reason is that the chemical space is vast, discrete and complex, and it is nontrivial to manipulate multiple properties of a molecule at the same time.

## 7 CONCLUSION

In this paper, we formulate a new task, molecule manipulation which allows interacting with molecule generative models, interpreting molecular space and generating molecules with smooth and monotonic property control. We present *MolSpacE*, a simple yet effective model-agnostic molecule manipulation method that achieves a high success in controlling and interpreting a variety of molecular properties. The interface illustrates the promising application of interactive molecule discovery. In the future, we plan to study how molecule manipulation can be done with unsupervised discovery.

Figure 6: A Real-time Interactive System Interface. Please refer to Appendix E demo video for interactive molecule discovery.

**Reproducibility Statement.** To increase the reproducibility of our paper, we provide detailed experiment setups in section 6.1 including datasets, baselines, implementation details. Additional experiments can be

found in Appendix C and D. We will put a link to our code in the comment when the discussion forum is open as suggested by the Author Guide. The designed interactive system along with a demo video to demonstrate how to use the system is shown in Appendix E.

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

# Appendix for
# "Interpreting Molecule Generative Models
# for Interactive Molecule Discovery"

## A    MOLECULE GENERATIVE MODELS

In Table 2, we summarize a list of representative molecule generative models, which span various types of deep generative models, including the type of generative models, the type of generation process and whether latent space is learned. We also provide the formulation for two types of deep generative models (VAE and Flow) in Section A that are very popular for molecule generation task.

Table 2: A summary of mainstream molecule generative models.

| Prior Work | Generative Model | Sequential | Latent Space |
|---|---|---|---|
| JT-VAE (Jin et al., 2018a) | VAE | ✓ | ✓ |
| CGVAE (Liu et al., 2018) | VAE | ✓ | ✓ |
| MRNN (Popova et al., 2019) | RNN | ✓ | |
| GraphNVP (Madhawa et al., 2019) | Flow | | ✓ |
| GCPN (You et al., 2018a) | RL | ✓ | |
| GraphAF (Shi et al., 2020) | Flow | ✓ | |
| MoFlow (Zang & Wang, 2020) | Flow | | ✓ |
| HierVAE (Jin et al., 2020) | VAE | ✓ | ✓ |
| GraphEBM (Liu et al., 2021) | EBM | | |
| GraphDF (Luo et al., 2021) | Flow | ✓ | |

### A.1    MOLECULE GENERATIVE MODEL FORMULATION

**VAE.** gets a lower bound (ELBO) for the data log probability by introducing a proposal distribution.

$$\begin{aligned}
\log p(x) &= \log \int_z p(x|z)p(z)dz \\
&\geq \log[\mathbb{E}_{q(z|x)}[p(x|z)] + \mathrm{KL}(q(z|x)||p(z))]
\end{aligned} \tag{16}$$

**Flow.** The key of Flow model is to design a invertible function with the following nice property:

$$\begin{aligned}
z_0 &\sim p_0(z_0) \\
z_i &= f_i(z_{i-1}) \\
z_{i-1} &= f_i^{-1}(z_i) \\
p_i(z_i) &= p_{i-1}(z_{i-1})\left|\det\frac{df_i^{-1}}{dz_i}\right| = p_{i-1}(f_i^{-1}(z_i))\left|\det\frac{df_i^{-1}}{dz_i}\right|,
\end{aligned} \tag{17}$$

where $f_i$ is invertible function. To be more concrete, we can take $z_0$ as some tractable noise distribution, like Gaussian distribution, and repeating this for $K$ steps will lead to the data distribution, *i.e.*,:

$$x = z_K = f_K \circ f_{K-1} \circ ... \circ f_1(z_0).$$

Thus, the log likelihood of the data is as follows:

$$\begin{aligned}
\log p(x) &= \log p_K(z_K) \\
&= \log p_{K-1}(z_{K-1}) - \log\left|\det\frac{df_K}{dz_{K-1}}\right| \\
&= \log p_{K-2}(z_{K-2}) - \log\left|\det\frac{df_{K-1}}{dz_{K-2}}\right| - \log\left|\det\frac{df_K}{dz_{K-1}}\right| \\
&= ... \\
&= \log p_0(z_0) - \sum_{i=1}^{K} \log\left|\det\frac{df_i}{dz_{i-1}}\right|
\end{aligned} \tag{18}$$

# B  STUDY OF MOLECULAR PROPERTIES

**List of Molecular Properties.** In total, study 154 molecular properties from the open chemistry library RDKit[3]. They are MaxEStateIndex, MinEStateIndex, MaxAbsEStateIndex, MinAbsEStateIndex, qed, MolWt, HeavyAtomMolWt, ExactMolWt, MaxPartialCharge, MinPartialCharge, MaxAbsPartialCharge, MinAbsPartialCharge, FpDensityMorgan1, FpDensityMorgan2, FpDensityMorgan3, MWHI, MWLOW, CHGHI, CHGLO, LOGPHI, LOGPLOW, MRHI, MRLOW, BalabanJ, BertzCT, Chi0, Chi0n, Chi0v, Chi1, Chi1n, Chi1v, Chi2n, Chi2v, Chi3n, Chi3v, Chi4n, Chi4v, HallKierAlpha, Ipc, Kappa1, Kappa2, Kappa3, LabuteASA, VSA13, VSA14, TPSA, EState1, EState10, EState2, EState3, EState4, EState5, EState6, EState7, EState8, EState9, FractionCSP3, MolLogP, MolMR, NumValenceElectrons, NumRadicalElectrons, HeavyAtomCount, NHOHCount, NOCount, NumAliphaticCarbocycles, NumAliphaticHeterocycles, NumAliphaticRings, NumAromaticCarbocycles, NumAromaticHeterocycles, NumAromaticRings, NumHAcceptors, NumHDonors, NumHeteroatoms, NumRotatableBonds, NumSaturatedCarbocycles, NumSaturatedHeterocycles, NumSaturatedRings, RingCount, noTert, ArN, N, NH, COO2, noCOO, S, HOCCN, Imine, NH0, NH1, NH2, Ndealkylation1, Ndealkylation2, Nhpyrrole, SH, aldehyde, carbamate, halide, oxid, amide, amidine, aniline, methyl, azide, azo, barbitur, benzene, benzodiazepine, bicyclic, diazo, dihydropyridine, epoxide, ether, furan, guanido, halogen, hdrzine, hdrzone, imidazole, imide, isocyan, isothiocyan, ketone, Topliss, lactam, lactone, methoxy, morpholine, nitrile, nitro, arom, nonortho, nitroso, oxazole, oxime, hydroxylation, phenol, noOrthoHbond, acid, piperdine, piperzine, priamide, prisulfonamd, pyridine, quatN, sulfide, sulfonamd, sulfone, acetylene, tetrazole, thiazole, thiocyan, thiophene, alkane, urea.

Among the studied 154 molecular properties, **59 of them are continuous**: MaxEStateIndex, MinEStateIndex, MaxAbsEStateIndex, MinAbsEStateIndex, qed, MolWt, HeavyAtomMolWt, ExactMolWt, MaxPartialCharge, MinPartialCharge, MaxAbsPartialCharge, MinAbsPartialCharge, FpDensityMorgan1, FpDensityMorgan2, FpDensityMorgan3, MWHI, MWLOW, CHGHI, CHGLO, LOGPHI, LOGPLOW, MRHI, MRLOW, BalabanJ, BertzCT, Chi0, Chi0n, Chi0v, Chi1, Chi1n, Chi1v, Chi2n, Chi2v, Chi3n, Chi3v, Chi4n, Chi4v, HallKierAlpha, Ipc, Kappa1, Kappa2, Kappa3, LabuteASA, VSA13, VSA14, TPSA, EState1, EState10, EState2, EState3, EState4, EState5, EState6, EState7, EState8, EState9, FractionCSP3, MolLogP, MolMR, **95 of them are categorical**: NumValenceElectrons, NumRadicalElectrons, HeavyAtomCount, NHOHCount, NOCount, NumAliphaticCarbocycles, NumAliphaticHeterocycles, NumAliphaticRings, NumAromaticCarbocycles, NumAromaticHeterocycles, NumAromaticRings, NumHAcceptors, NumHDonors, NumHeteroatoms, NumRotatableBonds, NumSaturatedCarbocycles, NumSaturatedHeterocycles, NumSaturatedRings, RingCount, noTert, ArN, N, NH, COO2, noCOO, S, HOCCN, Imine, NH0, NH1, NH2, Ndealkylation1, Ndealkylation2, Nhpyrrole, SH, aldehyde, carbamate, halide, oxid, amide, amidine, aniline, methyl, azide, azo, barbitur, benzene, benzodiazepine, bicyclic, diazo, dihydropyridine, epoxide, ether, furan, guanido, halogen, hdrzine, hdrzone, imidazole, imide, isocyan, isothiocyan, ketone, Topliss, lactam, lactone, methoxy, morpholine, nitrile, nitro, arom, nonortho, nitroso, oxazole, oxime, hydroxylation, phenol, noOrthoHbond, acid, piperdine, piperzine, priamide, prisulfonamd, pyridine, quatN, sulfide, sulfonamd, sulfone, acetylene, tetrazole, thiazole, thiocyan, thiophene, alkane, urea.

However, not all of the molecular properties are varied in the three datasets. Specifically, **QM9** contains 22 frozen molecular properties, NumRadicalElectrons, EState10, S, SH, azide, azo, barbitur, benzodiazepine, diazo, hdrzine, hdrzone, isocyan, isothiocyan, nitroso, acid, prisulfonamd, sulfide, sulfonamd, sulfone, thiazole, thiocyan, thiophene, **ZINC** contains 2 frozen molecular properties, NumRadicalElectrons and prisulfonamd and **ChEMBL** contains only 1 frozen molecular properties, prisulfonamd.

**Inter-correlations of molecular properties.** In Fig. 7, we visualize the linear correlations between each pair of molecular properties across three datasets. From the heatmaps, we can observe that there are no linear correlations between half of the molecular properties, and similar patterns are observed in ZINC and ChEMBL datasets.

**Molecular Property Distributions.** We visualize 7 molecular property distributions reported in section 6 in Fig. 8. From there, we can observe that the property distribution may vary a lot in terms

---

[3]https://www.rdkit.org/docs/index.html

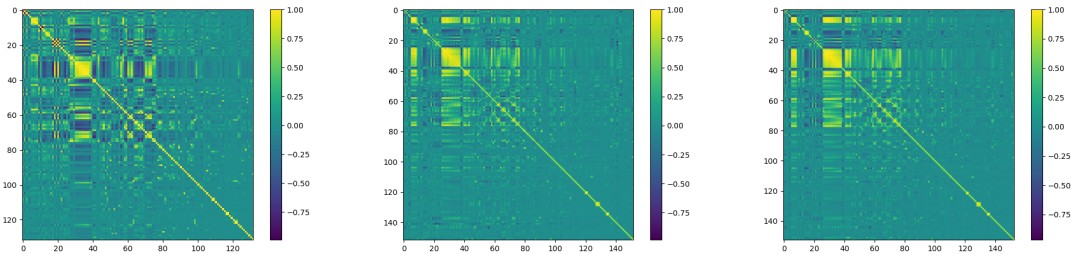

Figure 7: Inter-correlation heatmaps for studied molecular properties in QM9, ZINC and ChEMBL datasets.

of different datasets. Notably, the distributions of some properties, *e.g.*, QED, are very similar in ZINC and ChEMBL datasets, while some are quite different, *e.g.*, MolWt.

Figure 8: Property distributions of 7 randomly selected molecular properties on QM9, ZINC and ChEMBL datasets.

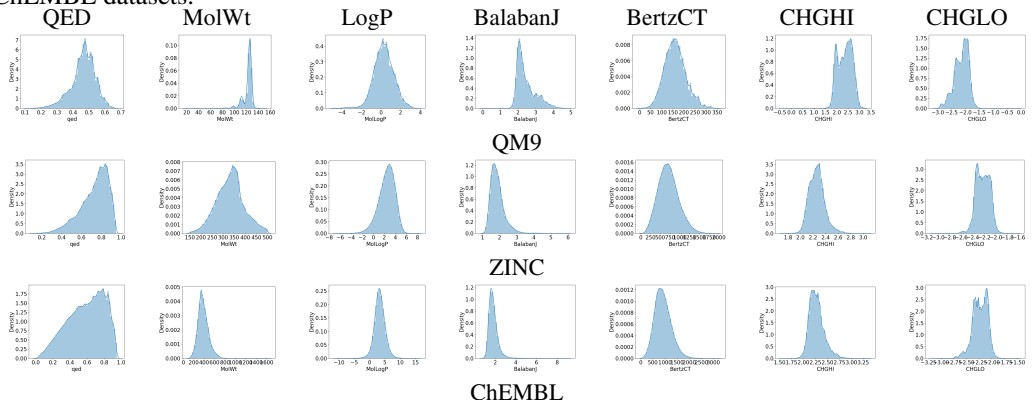

Table 3: Quantitative Evaluation of Disentanglement on Latent Space.

| Datasets | Models | Disentanglement | Compleness | Informativeness |
|---|---|---|---|---|
| QM9 | CGVAE | 0.15 | 0.49 | 0.79 |
| | MoFlow | **0.24** | **0.57** | **0.83** |
| | HierVAE | 0.13 | 0.27 | 0.75 |
| ZINC | CGVAE | 0.28 | 0.45 | **0.91** |
| | MoFlow | **0.40** | **0.62** | 0.87 |
| | HierVAE | 0.12 | 0.32 | 0.77 |
| ChEMBL | HierVAE | **0.14** | **0.41** | **0.81** |

## C  LATENT SPACE EVALUATION

To evaluate the quality of the learned latent space, we utilize three disentanglement evaluation metrics, disentanglement, completeness and informativeness (Eastwood & Williams, 2018). To be specific, disentanglement measures the degree to which each latent dimension controls at most one molecular property, completeness measures the degree to which each molecular property is governed by at most one latent dimension, and informativeness measures the prediction accuracy of molecular properties given the latent representation. From Table 3, we find MoFlow learns a better and more disentangled latent space than CGVAE and HierVAE. One possible reason is that MoFlow

(369) has a larger latent space than CGVAE (100) and HierVAE (32) since Flow restricts the latent size to be equal to the input size. Similarly, CGVAE ranks the second likely because its latent space size is larger than HierVAE.

## D MOLECULE MANIPULATION EXPERIMENTS

### D.1 MOLECULE GENRATION EVALUATION

We evaluate the **Validity**, **Novelty** and **Uniqueness** of the generated molecules as proposed in Kusner et al. (2017) in Table 4. We can observe that MolSpacE not only improves the success rate from the baseline methods, but also in general improves the validity, novelty, and uniqueness.

Besides, in Fig. 9, we also report the SSR curves of molecule manipulations over three models on QM9 and ZINC datasets with multiple manipulation ranges (distance in the latent space), $[-1, 1]$, $[-5, 5]$, $[-10, 10]$ and $[-20, 20]$. From the figure, we can observe that the trends in each of the curves remain still when the manipulation range changes. In general, either too large or too small range is not desired, we set it as a hyper-parameter and we observe that $[-1, 1]$ is a reasonably good default value. More experiments on molecule manipulation can be found in Appendix D.

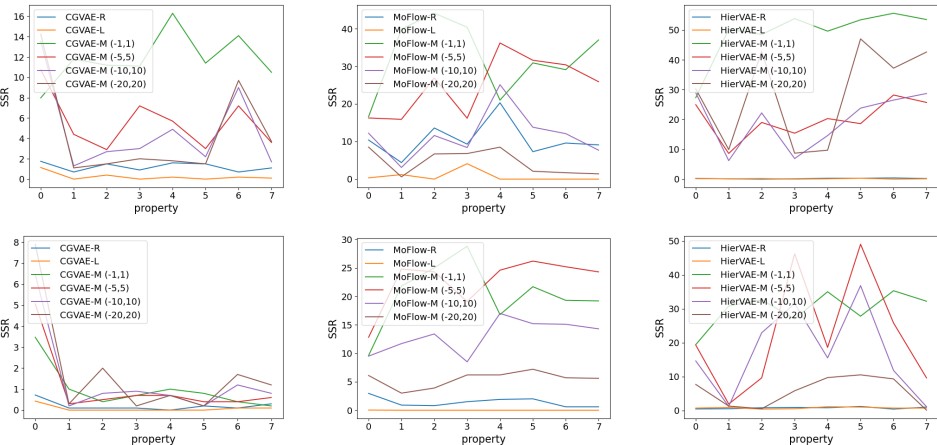

Figure 9: Molecule manipulation performance (average) with various manipulation ranges with three models on QM9 (top) and ZINC (bottom) datasets.

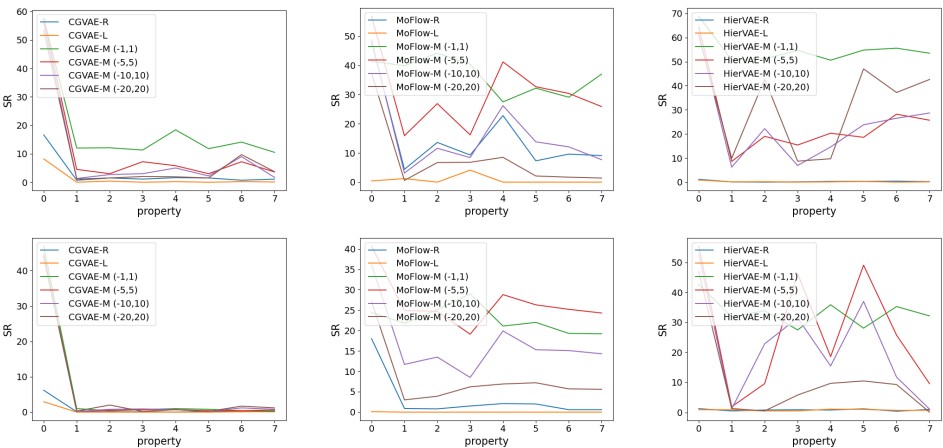

Figure 10: Molecule manipulation performance with various manipulation ranges with three models on QM9 (top) and ZINC (bottom) datasets (better seen in color).

## D.2 Molecule Manipulation Evaluation

In this section, we report detailed results for all manipulation ranges $[-1, 1]$, $[-5, 5]$, $[-10, 10]$, $[-20, 20]$ in terms of success rate and strict success rate in Table 5 and 5. Additionally, we visualize the SSR curves of molecule manipulations over three models on QM9 and ZINC in Fig. 10 and SR/SSR curves of molecule manipulation with HierVAE on ChEMBL datasets in Fig. 12. The manipulation visualization of CGVAE on QED, MolWt and LogP is provided in Fig. 13.

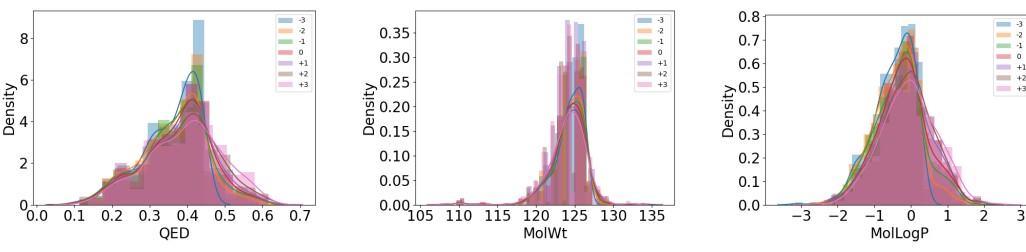

Figure 11: Visualization of Molecular property distribution shift while manipulating molecules with MoFlow on QM9 dataset (0 denotes the randomly sampled base molecule and $+x$ and $-x$ denote manipulation directions and steps).

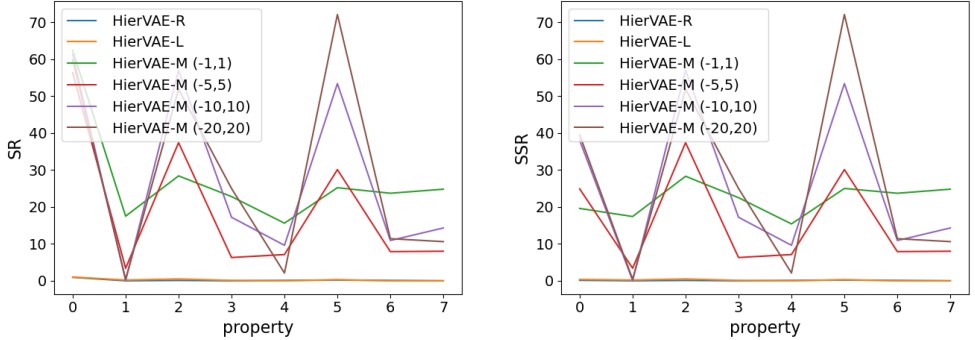

Figure 12: Molecule manipulation performance with various manipulation ranges with HierVAE on ChEMBL dataset (left SR, right SSR) (better seen in color).

## D.3 Molecule Manipulation Evaluation (Categorical)

To extensively demonstrate our results, we also randomly select four categorical molecular properties and show the results in terms of SR and SSR in Table 7 and 8, respectively. We can observe that the gaps between SR and SSR are much larger than the gaps in continuous molecular properties since it is easier for the method to generate molecules with same molecular property scores for the categorical data.

## E MolSpacE Demo

As shown in Fig. 6, we design an interactive real-time system for molecule manipulation, where the user can click random to randomly sample a molecule and freely select which model to interpret, which property to interpret, and tuning the slide bar manipulates the molecule accordingly. The demo video is anonymously provided at https://drive.google.com/drive/folders/1N036p_5OfvGZybgPJ3Vw1ONXHVepimSR?usp=sharing.

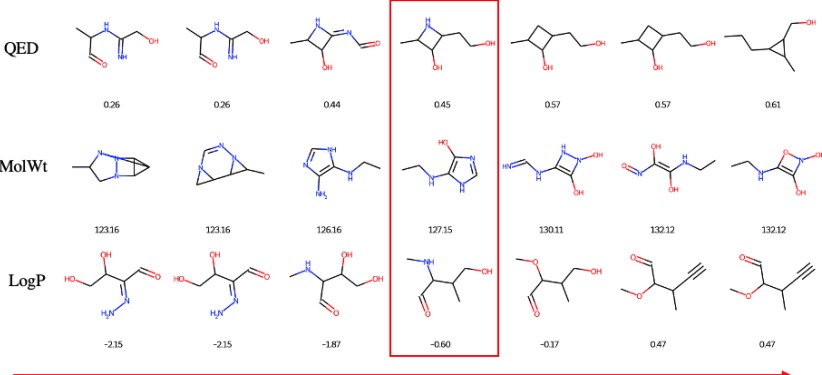

Figure 13: Manipulating QED, MolWt and LogP properties of sampled molecules with CGVAE model trained on QM9 dataset.

Table 4: Quantitative Evaluation of Latent Manipulation.

| Datasets | Models | Success (%) | Validity (%) | Novelty (%) | Uniqueness (%) |
|---|---|---|---|---|---|
| QM9 | CGVAE | N/A | 100.00% | 76.54% | 93.68% |
| | CGVAE-R (-1,1) | 16.64 | 100.00 | 99.55 | 13.69 |
| | CGVAE-L (-1,1) | 8.21 | 100.00 | 99.58 | 40.92 |
| | CGVAE-M (-1,1) | 57.59 | 100.00 | 73.81 | 56.60 |
| | CGVAE-M (-5,5) | 57.52 | 100.00 | 75.11 | 81.27 |
| | CGVAE-M (-10,10) | 55.71 | 100.00 | 76.79 | 89.06 |
| | CGVAE-M (-20,20) | 53.29 | 100.00 | 79.53 | 87.52 |
| QM9 | MoFlow | N/A | 100.00 | 98.23 | 98.27 |
| | MoFlow-R (-1,1) | 47.97 | 91.60 | 91.60 | 8.06 |
| | MoFlow-L (-1,1) | 0.41 | 40.75 | 40.75 | 9.32 |
| | MoFlow-M (-1,1) | 41.38 | 91.63 | 88.71 | 24.23 |
| | MoFlow-M (-5,5) | 56.84 | 91.93 | 89.13 | 51.21 |
| | MoFlow-M (-10,10) | 48.66 | 91.64 | 88.80 | 67.65 |
| | MoFlow-M (-20,20) | 37.25 | 90.15 | 87.27 | 78.42 |
| QM9 | HierVAE | N/A | 100.00 | 79.39 | 95.14 |
| | HierVAE-R (-1,1) | 1.16 | 100.00 | 84.53 | 28.91 |
| | HierVAE-L (-1,1) | 0.87 | 100.00 | 84.05 | 27.26 |
| | HierVAE-M (-1,1) | 68.84 | 100.00 | 79.66 | 34.81 |
| | HierVAE-M (-5,5) | 64.47 | 100.00 | 80.28 | 73.50 |
| | HierVAE-M (-10,10) | 61.61 | 100.00 | 81.29 | 78.77 |
| | HierVAE-M (-20,20) | 60.68 | 100.00 | 82.75 | 63.82 |
| ZINC | CGVAE | N/A | 100.00 | 99.99 | 98.51 |
| | CGVAE-R (-1,1) | 6.18 | 100.00 | 100.00 | 64.76 |
| | CGVAE-L (-1,1) | 2.94 | 100.00 | 100.00 | 82.26 |
| | CGVAE-M (-1,1) | 46.91 | 100.00 | 100.00 | 89.13 |
| | CGVAE-M (-5,5) | 44.65 | 100.00 | 99.99 | 95.17 |
| | CGVAE-M (-10,10) | 43.63 | 100.00 | 99.99 | 96.91 |
| | CGVAE-M (-20,20) | 44.10 | 100.00 | 99.99 | 96.83 |
| ZINC | MoFlow | N/A | 100.00 | 100.00 | 100.00 |
| | MoFlow-R (-1,1) | 18.00 | 69.98 | 69.98 | 29.04 |
| | MoFlow-L (-1,1) | 0.11 | 43.36 | 43.36 | 24.87 |
| | MoFlow-M (-1,1) | 24.43 | 71.26 | 71.26 | 15.82 |
| | MoFlow-M (-5,5) | 40.66 | 70.72 | 70.72 | 31.72 |
| | MoFlow-M (-10,10) | 36.05 | 70.43 | 70.43 | 44.03 |
| | MoFlow-M (-20,20) | 27.12 | 70.07 | 70.07 | 56.62 |
| ZINC | HierVAE | N/A | 100.00 | 100.00 | 3.27 |
| | HierVAE-R (-1,1) | 1.26 | 100.00 | 100.00 | 1.22 |
| | HierVAE-L (-1,1) | 0.91 | 100.00 | 100.00 | 2.43 |
| | HierVAE-M (-1,1) | 42.67 | 100.00 | 100.00 | 3.07 |
| | HierVAE-M (-5,5) | 55.41 | 100.00 | 99.99 | 4.85 |
| | HierVAE-M (-10,10) | 52.68 | 100.00 | 99.99 | 6.26 |
| | HierVAE-M (-20,20) | 46.99 | 100.00 | 99.99 | 7.85 |
| ChEMBL | HierVAE | N/A | 100.00 | 94.03 | 99.45 |
| | HierVAE-R (-1,1) | 1.16 | 100.00 | 84.53 | 28.91 |
| | HierVAE-L (-1,1) | 1.02 | 100.00 | 93.00 | 55.09 |
| | HierVAE-M (-1,1) | 62.38 | 100.00 | 94.24 | 56.58 |
| | HierVAE-M (-5,5) | 56.33 | 100.00 | 92.65 | 92.51 |
| | HierVAE-M (-10,10) | 61.38 | 100.00 | 91.39 | 89.72 |
| | HierVAE-M (-20,20) | 59.83 | 100.00 | 91.49 | 75.77 |

Table 5: Quantitative Evaluation of Molecule Manipulation over a variety of molecular properties (numbers reported are *success rate* in %, -R denotes model with random manipulation, -L denotes model with largest range manipulation, -M denotes model with MolSpacE).

| Datasets | Models | Avg. | QED | MW | logP | BalabanJ | BertzCT | CHGHI | CHGLO |
|---|---|---|---|---|---|---|---|---|---|
| QM9 | CGVAE-R | 16.64 | 0.70 | 1.50 | 1.10 | 1.60 | 1.50 | 0.70 | 1.10 |
| | CGVAE-L | 8.21 | 0.00 | 0.40 | 0.00 | 0.20 | 0.00 | 0.20 | 0.10 |
| | CGVAE-M (-1,1) | 57.59 | 12.00 | 12.10 | 11.30 | 18.40 | 11.80 | 14.10 | 10.50 |
| | CGVAE-M (-5,5) | 57.52 | 4.50 | 3.00 | 7.20 | 5.80 | 3.00 | 7.20 | 3.60 |
| | CGVAE-M (-10,10) | 55.71 | 1.30 | 2.70 | 3.00 | 5.00 | 2.20 | 9.00 | 1.70 |
| | CGVAE-M (-20,20) | 53.29 | 1.10 | 1.50 | 2.00 | 1.90 | 1.50 | 9.70 | 3.70 |
| | MoFlow-R | 47.97 | 4.40 | 13.60 | 9.30 | 22.80 | 7.30 | 9.60 | 9.10 |
| | MoFlow-L | 0.41 | 1.20 | 0.00 | 4.10 | 0.00 | 0.00 | 0.00 | 0.00 |
| | MoFlow-M (-1,1) | 41.38 | 39.90 | 44.20 | 40.40 | 27.50 | 32.20 | 29.10 | 37.00 |
| | MoFlow-M (-5,5) | 56.84 | 15.90 | 26.90 | 16.20 | 41.20 | 32.70 | 30.40 | 25.90 |
| | MoFlow-M (-10,10) | 48.66 | 3.10 | 11.60 | 8.40 | 26.20 | 13.80 | 12.10 | 7.70 |
| | MoFlow-M (-20,20) | 37.25 | 0.60 | 6.70 | 6.80 | 8.50 | 2.10 | 1.70 | 1.40 |
| | HierVAE-R | 1.16 | 0.10 | 0.00 | 0.10 | 0.30 | 0.30 | 0.40 | 0.20 |
| | HierVAE-L | 0.87 | 1.70 | 2.00 | 0.00 | 0.00 | | | |
| | HierVAE-M (-1,1) | 68.84 | 50.70 | 51.20 | 54.70 | 50.60 | 54.80 | 55.60 | 53.50 |
| | HierVAE-M (-5,5) | 64.47 | 8.60 | 19.00 | 15.40 | 20.30 | 18.60 | 28.20 | 25.70 |
| | HierVAE-M (-10,10) | 61.61 | 6.20 | 22.20 | 6.90 | 14.60 | 23.80 | 26.50 | 28.70 |
| | HierVAE-M (-20,20) | 60.68 | 9.90 | 42.80 | 8.70 | 9.70 | 47.00 | 37.20 | 42.60 |
| ZINC | CGVAE-R | 6.18 | 0.10 | 0.10 | 0.10 | 0.00 | 0.20 | 0.10 | 0.30 |
| | CGVAE-L | 2.94 | 0.00 | 0.00 | 0.00 | 0.00 | 0.00 | 0.10 | 0.10 |
| | CGVAE-M (-1,1) | 46.91 | 1.00 | 0.40 | 0.70 | 1.00 | 0.80 | 0.40 | 0.20 |
| | CGVAE-M (-5,5) | 44.65 | 0.30 | 0.50 | 0.70 | 0.70 | 0.40 | 0.40 | 0.60 |
| | CGVAE-M (-10,10) | 43.63 | 0.20 | 0.80 | 0.90 | 0.70 | 0.20 | 1.20 | 0.80 |
| | CGVAE-M (-20,20) | 44.10 | 0.30 | 2.00 | 0.20 | 0.70 | 0.20 | 1.70 | 1.20 |
| | MoFlow-R | 18.00 | 0.90 | 0.80 | 1.50 | 2.10 | 2.00 | 0.60 | 0.60 |
| | MoFlow-L | 0.11 | 0.00 | 0.00 | 0.00 | 0.00 | 0.00 | 0.00 | 0.00 |
| | MoFlow-M (-1,1) | 24.43 | 21.80 | 25.10 | 29.00 | 21.10 | 22.00 | 19.30 | 19.20 |
| | MoFlow-M (-5,5) | 40.66 | 24.90 | 24.70 | 19.10 | 28.80 | 26.30 | 25.20 | 24.30 |
| | MoFlow-M (-10,10) | 36.05 | 11.70 | 13.50 | 8.50 | 19.90 | 15.30 | 15.10 | 14.30 |
| | MoFlow-M (-20,20) | 27.12 | 3.00 | 3.90 | 6.20 | 6.90 | 7.20 | 5.70 | 5.60 |
| | HierVAE-R | 1.26 | 0.50 | 0.80 | 0.90 | 0.80 | 1.20 | 0.40 | 1.00 |
| | HierVAE-L | 0.91 | 0.90 | 0.50 | 0.50 | 1.10 | 1.00 | 0.70 | 0.70 |
| | HierVAE-M (-1,1) | 42.67 | 31.90 | 33.90 | 27.50 | 35.90 | 28.10 | 35.30 | 32.20 |
| | HierVAE-M (-5,5) | 55.41 | 1.90 | 9.60 | 46.30 | 18.60 | 49.10 | 25.80 | 9.60 |
| | HierVAE-M (-10,10) | 52.68 | 1.20 | 22.90 | 31.50 | 15.50 | 37.00 | 11.80 | 1.10 |
| | HierVAE-M (-20,20) | 46.99 | 1.30 | 0.50 | 5.80 | 9.70 | 10.50 | 9.30 | 0.10 |
| ChEMBL | HierVAE-R | 0.93 | 0.00 | 0.10 | 0.00 | 0.10 | 0.20 | 0.10 | 0.00 |
| | HierVAE-L | 1.02 | 0.20 | 0.50 | 0.10 | 0.00 | 0.30 | 0.00 | 0.00 |
| | HierVAE-M (-1,1) | 62.38 | 17.50 | 28.40 | 22.80 | 15.60 | 25.20 | 23.70 | 24.80 |
| | HierVAE-M (-5,5) | 56.33 | 3.40 | 37.40 | 6.30 | 7.10 | 30.10 | 7.90 | 8.00 |
| | HierVAE-M (-10,10) | 61.38 | 0.10 | 57.00 | 17.20 | 9.60 | 53.40 | 10.90 | 14.30 |
| | HierVAE-M (-20,20) | 59.83 | 0.40 | 51.90 | 25.00 | 2.10 | 72.10 | 11.40 | 10.60 |

Table 6: Quantitative Evaluation of Molecule Manipulation over a variety of molecular properties (numbers reported are *strict success rate* in %, -R denotes model with random manipulation, -L denotes model with largest range manipulation, -M denotes model with MolSpacE).

| Datasets | Models | Avg. | QED | MW | logP | BalabanJ | BertzCT | CHGHI | CHGLO |
|---|---|---|---|---|---|---|---|---|---|
| | CGVAE-R | 1.75 | 0.70 | 1.50 | 0.90 | 1.60 | 1.50 | 0.70 | 1.10 |
| | CGVAE-L | 1.14 | 0.00 | 0.40 | 0.00 | 0.20 | 0.00 | 0.20 | 0.10 |
| | CGVAE-M (-1,1) | 7.98 | 11.80 | 11.20 | 11.10 | 16.30 | 11.40 | 14.10 | 10.50 |
| | CGVAE-M (-5,5) | 10.75 | 4.40 | 2.90 | 7.20 | 5.70 | 3.00 | 7.20 | 3.60 |
| | CGVAE-M (-10,10) | 13.54 | 1.30 | 2.70 | 3.00 | 4.90 | 2.20 | 9.00 | 1.70 |
| | CGVAE-M (-20,20) | 14.25 | 1.10 | 1.50 | 2.00 | 1.80 | 1.50 | 9.70 | 3.70 |
| | MoFlow-R | 10.40 | 4.40 | 13.60 | 9.30 | 20.30 | 7.30 | 9.60 | 9.10 |
| | MoFlow-L | 0.35 | 1.20 | 0.00 | 4.10 | 0.00 | 0.00 | 0.00 | 0.00 |
| QM9 | MoFlow-M (-1,1) | 16.62 | 39.90 | 44.10 | 40.40 | 21.00 | 30.90 | 29.10 | 37.00 |
| | MoFlow-M (-5,5) | 16.26 | 15.90 | 26.90 | 16.20 | 36.20 | 31.60 | 30.40 | 25.90 |
| | MoFlow-M (-10,10) | 12.21 | 3.10 | 11.60 | 8.40 | 25.10 | 13.80 | 12.10 | 7.70 |
| | MoFlow-M (-20,20) | 8.51 | 0.60 | 6.70 | 6.80 | 8.50 | 2.10 | 1.70 | 1.40 |
| | HierVAE-R | 0.19 | 0.10 | 0.00 | 0.10 | 0.30 | 0.30 | 0.40 | 0.20 |
| | HierVAE-L | 0.24 | 0.10 | 0.20 | 0.00 | 0.10 | 0.30 | 0.00 | 0.10 |
| | HierVAE-M (-1,1) | 27.30 | 50.20 | 48.50 | 53.80 | 49.60 | 53.40 | 55.60 | 53.50 |
| | HierVAE-M (-5,5) | 24.96 | 8.60 | 19.00 | 15.40 | 20.30 | 18.60 | 28.20 | 25.70 |
| | HierVAE-M (-10,10) | 28.97 | 6.20 | 22.20 | 6.90 | 14.60 | 23.80 | 26.50 | 28.70 |
| | HierVAE-M (-20,20) | 30.28 | 9.90 | 42.80 | 8.70 | 9.70 | 47.00 | 37.20 | 42.60 |
| | CGVAE-R | 0.72 | 0.10 | 0.10 | 0.10 | 0.00 | 0.20 | 0.10 | 0.30 |
| | CGVAE-L | 0.43 | 0.00 | 0.00 | 0.00 | 0.00 | 0.00 | 0.10 | 0.10 |
| | CGVAE-M (-1,1) | 3.47 | 1.00 | 0.40 | 0.70 | 1.00 | 0.80 | 0.40 | 0.20 |
| | CGVAE-M (-5,5) | 5.06 | 0.30 | 0.50 | 0.70 | 0.70 | 0.40 | 0.40 | 0.60 |
| | CGVAE-M (-10,10) | 6.46 | 0.20 | 0.80 | 0.90 | 0.70 | 0.20 | 1.20 | 0.80 |
| | CGVAE-M (-20,20) | 7.90 | 0.30 | 2.00 | 0.20 | 0.70 | 0.20 | 1.70 | 1.20 |
| | MoFlow-R | 2.96 | 0.90 | 0.80 | 1.50 | 1.90 | 2.00 | 0.60 | 0.60 |
| | MoFlow-L | 0.03 | 0.00 | 0.00 | 0.00 | 0.00 | 0.00 | 0.00 | 0.00 |
| ZINC | MoFlow-M (-1,1) | 9.61 | 21.80 | 25.00 | 28.80 | 16.80 | 21.70 | 19.30 | 19.20 |
| | MoFlow-M (-5,5) | 12.81 | 24.80 | 24.30 | 19.10 | 24.60 | 26.20 | 25.20 | 24.30 |
| | MoFlow-M (-10,10) | 9.50 | 11.70 | 13.40 | 8.50 | 17.00 | 15.20 | 15.10 | 14.30 |
| | MoFlow-M (-20,20) | 6.10 | 3.00 | 3.90 | 6.20 | 6.20 | 7.20 | 5.70 | 5.60 |
| | HierVAE-R | 0.45 | 0.50 | 0.80 | 0.90 | 0.80 | 1.20 | 0.40 | 1.00 |
| | HierVAE-L | 0.70 | 0.90 | 0.40 | 0.50 | 1.10 | 1.00 | 0.70 | 0.70 |
| | HierVAE-M (-1,1) | 19.40 | 30.80 | 32.60 | 27.00 | 35.00 | 27.80 | 35.30 | 32.20 |
| | HierVAE-M (-5,5) | 19.29 | 1.90 | 9.60 | 46.20 | 18.60 | 49.00 | 25.80 | 9.60 |
| | HierVAE-M (-10,10) | 14.61 | 1.20 | 22.90 | 31.50 | 15.50 | 36.80 | 11.80 | 1.10 |
| | HierVAE-M (-20,20) | 7.68 | 1.30 | 0.50 | 5.80 | 9.70 | 10.50 | 9.30 | 0.10 |
| | HierVAE-R | 0.13 | 0.00 | 0.10 | 0.00 | 0.10 | 0.20 | 0.10 | 0.00 |
| | HierVAE-L | 0.36 | 0.20 | 0.50 | 0.10 | 0.00 | 0.30 | 0.00 | 0.00 |
| ChEMBL | HierVAE-M (-1,1) | 19.59 | 17.40 | 28.30 | 22.50 | 15.40 | 25.00 | 23.70 | 24.80 |
| | HierVAE-M (-5,5) | 24.89 | 3.40 | 37.40 | 6.30 | 7.10 | 30.10 | 7.90 | 8.00 |
| | HierVAE-M (-10,10) | 38.01 | 0.10 | 57.00 | 17.20 | 9.60 | 53.40 | 10.90 | 14.30 |
| | HierVAE-M (-20,20) | 39.47 | 0.40 | 51.90 | 25.00 | 2.10 | 72.10 | 11.40 | 10.60 |

Table 7: Quantitative Evaluation of Molecule Manipulation over four categorical molecular properties (numbers reported are *success rate* in %, -R denotes model with random manipulation, -L denotes model with largest range manipulation, -M denotes model with MolSpacE).

| Datasets | Models | Avg. | acetylene | tetrazole | thiazole | thiocyan |
|---|---|---|---|---|---|---|
| QM9 | CGVAE-R | 16.64 | 25.40 | 29.60 | 29.60 | 29.60 |
| | CGVAE-L | 8.21 | 12.80 | 15.90 | 13.90 | 13.40 |
| | CGVAE-M (-1,1) | 57.59 | 85.10 | 95.60 | 95.40 | 93.10 |
| | CGVAE-M (-5,5) | 57.52 | 85.80 | 99.80 | 100.00 | 99.90 |
| | CGVAE-M (-10,10) | 55.71 | 78.10 | 100.00 | 100.00 | 99.90 |
| | CGVAE-M (-20,20) | 53.29 | 65.90 | 99.90 | 100.00 | 100.00 |
| | MoFlow-R | 47.97 | 73.80 | 75.90 | 76.20 | 76.90 |
| | MoFlow-L | 0.41 | 0.00 | 0.00 | 0.00 | 0.00 |
| | MoFlow-M (-1,1) | 41.38 | 33.50 | 45.70 | 31.60 | 36.10 |
| | MoFlow-M (-5,5) | 56.84 | 71.70 | 78.30 | 74.60 | 73.90 |
| | MoFlow-M (-10,10) | 48.66 | 72.90 | 74.30 | 77.50 | 74.70 |
| | MoFlow-M (-20,20) | 37.25 | 61.20 | 58.70 | 68.90 | 64.50 |
| | HierVAE-R | 1.16 | 1.70 | 2.00 | 2.00 | 2.00 |
| | HierVAE-L | 0.87 | 1.70 | 2.00 | 0.00 | 0.00 |
| | HierVAE-M (-1,1) | 68.84 | 77.00 | 83.00 | 74.00 | 78.10 |
| | HierVAE-M (-5,5) | 64.47 | 91.10 | 99.50 | 99.90 | 100.00 |
| | HierVAE-M (-10,10) | 61.61 | 88.30 | 98.50 | 100.00 | 100.00 |
| | HierVAE-M (-20,20) | 60.68 | 83.70 | 95.50 | 100.00 | 100.00 |
| ZINC | CGVAE-R | 6.18 | 14.20 | 14.20 | 13.60 | 14.20 |
| | CGVAE-L | 2.94 | 7.10 | 7.10 | 6.60 | 7.00 |
| | CGVAE-M (-1,1) | 46.91 | 97.30 | 99.00 | 93.90 | 99.60 |
| | CGVAE-M (-5,5) | 44.65 | 97.60 | 99.30 | 95.50 | 99.70 |
| | CGVAE-M (-10,10) | 43.63 | 97.40 | 99.60 | 95.70 | 99.70 |
| | CGVAE-M (-20,20) | 44.10 | 97.90 | 99.50 | 95.10 | 99.80 |
| | MoFlow-R | 18.00 | 31.80 | 31.20 | 31.60 | 34.50 |
| | MoFlow-L | 0.11 | 0.30 | 0.00 | 0.50 | 0.00 |
| | MoFlow-M (-1,1) | 24.43 | 23.20 | 23.10 | 22.20 | 22.60 |
| | MoFlow-M (-5,5) | 40.66 | 49.30 | 50.60 | 48.50 | 47.50 |
| | MoFlow-M (-10,10) | 36.05 | 51.50 | 53.60 | 54.30 | 52.50 |
| | MoFlow-M (-20,20) | 27.12 | 46.90 | 46.30 | 46.30 | 45.40 |
| | HierVAE-R | 1.26 | 1.90 | 1.70 | 1.40 | 1.50 |
| | HierVAE-L | 0.91 | 0.00 | 2.00 | 1.60 | 0.00 |
| | HierVAE-M (-1,1) | 42.67 | 19.90 | 57.20 | 35.60 | 50.40 |
| | HierVAE-M (-5,5) | 55.41 | 66.40 | 97.50 | 84.50 | 93.10 |
| | HierVAE-M (-10,10) | 52.68 | 95.60 | 99.60 | 98.70 | 99.90 |
| | HierVAE-M (-20,20) | 46.99 | 100.00 | 96.70 | 100.00 | 100.00 |
| ChEMBL | HierVAE-R | 0.13 | 0.00 | 0.00 | 0.00 | 0.00 |
| | HierVAE-L | 0.36 | 0.10 | 0.00 | 0.00 | 0.00 |
| | HierVAE-M (-1,1) | 62.38 | 93.20 | 92.60 | 94.30 | 92.80 |
| | HierVAE-M (-5,5) | 56.33 | 98.80 | 95.30 | 90.30 | 100.00 |
| | HierVAE-M (-10,10) | 61.38 | 95.20 | 87.30 | 81.20 | 99.90 |
| | HierVAE-M (-20,20) | 59.83 | 87.50 | 74.70 | 74.00 | 99.90 |

Table 8: Quantitative Evaluation of Molecule Manipulation over four categorical molecular properties (numbers reported are *strict success rate* in %, -R denotes model with random manipulation, -L denotes model with largest range manipulation, -M denotes model with MolSpacE).

| Datasets | Models | Avg. | acetylene | tetrazole | thiazole | thiocyan |
|----------|--------|------|-----------|-----------|----------|----------|
| QM9 | CGVAE-R | 1.75 | 2.30 | 0.10 | 0.00 | 0.00 |
| | CGVAE-L | 1.14 | 1.20 | 0.10 | 0.00 | 0.00 |
| | CGVAE-M (-1,1) | 7.98 | 5.00 | 0.00 | 0.00 | 0.00 |
| | CGVAE-M (-5,5) | 10.75 | 17.00 | 0.00 | 0.00 | 0.00 |
| | CGVAE-M (-10,10) | 13.54 | 20.00 | 0.00 | 0.00 | 0.00 |
| | CGVAE-M (-20,20) | 14.25 | 13.70 | 0.20 | 0.00 | 0.00 |
| | MoFlow-R | 10.40 | 7.20 | 0.70 | 0.00 | 0.00 |
| | MoFlow-L | 0.35 | 0.00 | 0.00 | 0.00 | 0.00 |
| | MoFlow-M (-1,1) | 16.62 | 2.40 | 0.20 | 0.00 | 0.00 |
| | MoFlow-M (-5,5) | 16.26 | 7.70 | 0.30 | 0.00 | 0.00 |
| | MoFlow-M (-10,10) | 12.21 | 9.20 | 0.50 | 0.00 | 0.00 |
| | MoFlow-M (-20,20) | 8.51 | 4.40 | 0.30 | 0.00 | 0.00 |
| | HierVAE-R | 0.19 | 0.10 | 0.00 | 0.00 | 0.00 |
| | HierVAE-L | 0.24 | 0.70 | 0.10 | 0.00 | 0.00 |
| | HierVAE-M (-1,1) | 27.30 | 12.80 | 0.20 | 0.00 | 0.00 |
| | HierVAE-M (-5,5) | 24.96 | 54.70 | 1.50 | 0.00 | 0.00 |
| | HierVAE-M (-10,10) | 28.97 | 82.30 | 4.40 | 0.00 | 0.00 |
| | HierVAE-M (-20,20) | 30.28 | 82.70 | 2.90 | 0.00 | 0.00 |
| ZINC | CGVAE-R | 0.72 | 0.20 | 0.00 | 0.10 | 0.00 |
| | CGVAE-L | 0.43 | 1.50 | 0.10 | 0.40 | 0.00 |
| | CGVAE-M (-1,1) | 3.47 | 0.80 | 0.10 | 1.80 | 0.00 |
| | CGVAE-M (-5,5) | 5.06 | 0.60 | 0.20 | 2.50 | 0.00 |
| | CGVAE-M (-10,10) | 6.46 | 1.00 | 0.10 | 2.60 | 0.10 |
| | CGVAE-M (-20,20) | 7.90 | 1.70 | 0.40 | 4.90 | 0.100 |
| | MoFlow-R | 2.96 | 0.10 | 0.00 | 0.00 | 0.00 |
| | MoFlow-L | 0.03 | 0.10 | 0.00 | 0.00 | 0.00 |
| | MoFlow-M (-1,1) | 9.61 | 0.00 | 0.00 | 0.00 | 0.00 |
| | MoFlow-M (-5,5) | 12.81 | 0.10 | 0.00 | 0.00 | 0.00 |
| | MoFlow-M (-10,10) | 9.50 | 0.00 | 0.00 | 0.00 | 0.00 |
| | MoFlow-M (-20,20) | 6.10 | 0.10 | 0.00 | 0.00 | 0.00 |
| | HierVAE-R | 0.45 | 0.00 | 0.00 | 0.00 | 0.00 |
| | HierVAE-L | 0.70 | 0.00 | 0.20 | 0.00 | 0.00 |
| | HierVAE-M (-1,1) | 19.40 | 0.00 | 0.00 | 0.00 | 0.00 |
| | HierVAE-M (-5,5) | 19.29 | 0.00 | 0.80 | 0.10 | 0.00 |
| | HierVAE-M (-10,10) | 14.61 | 0.00 | 2.90 | 0.10 | 0.00 |
| | HierVAE-M (-20,20) | 7.68 | 0.00 | 1.60 | 0.00 | 0.00 |
| ChEMBL | HierVAE-R | 0.13 | 0.00 | 0.00 | 0.00 | 0.00 |
| | HierVAE-L | 0.36 | 0.10 | 0.00 | 0.00 | 0.00 |
| | HierVAE-M (-1,1) | 19.59 | 1.20 | 2.20 | 2.90 | 0.10 |
| | HierVAE-M (-5,5) | 24.89 | 7.30 | 11.00 | 16.30 | 0.00 |
| | HierVAE-M (-10,10) | 38.01 | 21.10 | 21.50 | 28.50 | 0.00 |
| | HierVAE-M (-20,20) | 39.47 | 16.90 | 9.00 | 42.40 | 0.00 |

