# OpenReview forum: "Interpreting Molecule Generative Models for Interactive Molecule Discovery"
_ICLR.cc/2022/Conference — ICLR 2022 Submitted_

### Official Review · Reviewer_Pgt6 · 2021-11-02

**Correctness:** 3
**Technical Novelty And Significance:** 2
**Empirical Novelty And Significance:** 2
**Recommendation:** 3
**Confidence:** 4

**Main Review:**

My first concern about this study is that there are many methods that have simlar concepts (model-agnostic and does not require re-training of the molecule generative model). Some examples are list below. They commonly perform optimization on the latent space toward desired properties without modifying the molecular generative model. Also, they are model-agnostic, applicable to any molecule generative model. I'm wondering if the proposed method is superior to or is advantages in any aspect over existing methods.

Gómez-Bombarelli, R., Wei, J. N., Duvenaud, D., Hernández-Lobato, J. M., Sánchez-Lengeling, B., Sheberla, D., ... & Aspuru-Guzik, A. (2018). Automatic chemical design using a data-driven continuous representation of molecules. ACS central science, 4(2), 268-276.
Griffiths, R. R., & Hernández-Lobato, J. M. (2020). Constrained Bayesian optimization for automatic chemical design using variational autoencoders. Chemical science, 11(2), 577-586.
Notin, P., Hernández-Lobato, J. M., & Gal, Y. (2021). Improving black-box optimization in VAE latent space using decoder uncertainty. arXiv preprint arXiv:2107.00096.
--> bayesian optimization on the latent space to manipulate molecule.
Kwon, Y., Kang, S., Choi, Y. S., & Kim, I. (2021). Evolutionary design of molecules based on deep learning and a genetic algorithm. Scientific reports, 11(1), 1-11.
--> genetic algorithm on the latent space to manipulate molecule.
Winter, R., Montanari, F., Steffen, A., Briem, H., Noé, F., & Clevert, D. A. (2019). Efficient multi-objective molecular optimization in a continuous latent space. Chemical science, 10(34), 8016-8024.
--> particle swarm optimization on the latent space to manipulate molecule.

My second concern is whether the proposed method provides "interpretability". While the authors metioned theyir method is effective on interpreting molecular generative models, the results the authors presented are gradual modifications of a query molecule and their respective properties, just like other related studies. I don't see any particually "interpretable" stuffs.

**Summary Of The Paper:**

This paper presents a method named "MolSpaceExplorer" that explores the latent space of a molecule generative model to continuously optimize molecules toward despred properties.They identify latent directions by constructing a serapation boundary (hyperplane) on the latent space and use the directions to improve latent vectors, which are then fed into the generatove model to produce new molecules with desired properties. The authors also developed an interface for interactive molecular discovery. The authors mentioned that the main contributions of this study are 1) this method is model-agnostic thus applicable to any molecule generative model 2) it does not require any retraining of the molecule generative model.




**Summary Of The Review:**

I think this paper does not demonstrate any significant contributions compared to existing studies.

---

### Official Review · Reviewer_bacV · 2021-11-02

**Correctness:** 3
**Technical Novelty And Significance:** 2
**Empirical Novelty And Significance:** 2
**Recommendation:** 3
**Confidence:** 5

**Main Review:**

## Overview

This paper addresses an important problem which few prior works have addressed before in the literature. Their rough approach is reasonable. However, I had a lot of questions and doubts, some major and some minor, which I will discuss below. I will use **(major+/-)** and **(minor+/-)** to denote issues which are extremely and moderately important to me, respectively.

## Problem definition

I am sympathetic to your problem setting of "continuously" tuning the properties of molecules **(minor+)**, as this could be used to make a lot of practical tools (such as the interesting demo you provide in the end). Roughly, given a starting molecule, you should be able to produce another similar molecule but with slightly higher or lower property values. I applaud your efforts to try to define this more precisely in section 4, but I think your definition has several significant problems **(major-)**:

- there seem to be no constraints on the molecules outputted (e.g. a similarity constraint to the starting molecule). Without this, your metrics could be easily exploited by having a sorted list of ~1000 molecules and associated property values, and given a starting molecule, output in order all molecules above/below the starting molecule's property value. A similarity constraint would remove this exploit, for example (although there are certainly other ways to do this).
- I think the monotonicity conditions in your SR and SSR are too strict, given the high degree of randomness in generative models. By your metrics, a property sequence of [1, 1, 1, 1.00001, 1.001, 1.001] would satisfy both SR and SSR, while [-3, -2, -1, 0.01, 0, 1, 2, 3] would not, even though the second covers a much broader property range and to me seems more in line with what practitioners would want. I acknowledge that this criticism is subjective.
- I object to the use of the word "continuous" in your definitions, which you use several times. All molecular outputs are discrete, and many of the properties you manipulate are integer valued. In the strict technical sense of the word, nothing you do has continuity, although it does match the non-technical usage of the word. Perhaps you could say "step-wise" or "monotonically" instead?

I acknowledge that creating a good definition of this is hard and I don't claim to have a great definition myself. The best rough suggestion I can think of would be to define a manipulation in $k$ steps as a sequence of molecules $m_1, \ldots, m_k$ such that $|f_P(m_i) - f_P(m_{i+1})| < \epsilon \forall i$ for some reasonable $\epsilon$, and that $\text{sim}(m_i, m_{i+1}) > \delta \forall i$ for some similarity metric. Given this, I would produce a  long sequence of molecules using your method, until the first instance where the conditions are violated and it ceases to be a valid manipulation. You could then score each manipulation based on its length (longer is better) and the magnitude of the property difference between the starting and ending molecules (larger is better). Which exact scoring function to use is unclear, as well as the appropriate value of $\epsilon, \delta$, but hopefully you get what I mean from this sketch.

## Related work

Your paper is closely related to a variety of existing works on molecule generation. While you have many important citations, I think that you are missing many others **(minor-)**:

- Bombarelli et al., 2018 should really be credited with the invention of BO in the latent space of generative models, not Jin et al
- a variety of other works which do molecule optimization [1-7]
- work on graph to graph translation, e.g. [8] which is very similar to spirit to yours
- work on conditional generative modelling for molecules (e.g [9]) which could also be used for your problem setting

There are also many other relevant works, especially in computer vision, which aim to modify images by manipulating latent vectors. As far as I could tell, no such analogous methods are mentioned in your paper. [10-13] are some starting references for you. Many of these papers propose methods similar enough to yours that you should discuss how the method you propose is different **(minor-)**.

## Your proposed method (section 5)

### A questionable argument in section 5.1

I was not sympathetic to your argument in section 5.1 **(minor-)**. While yes, molecules with similar structures tend to cluster together in latent space, and structures determine properties, it _does not follow_ that molecules with similar properties will cluster together.
This is because of _activity cliffs_, a common and well-understood phenomenon where molecules which are very similar can have very different properties [14]. They are caused by many things in biology having cut-offs or other non-linear effects (e.g. if a molecule is a little bit too big it can't fit into a receptor and will have 0 activity). Reading this section it felt like the authors were not aware of this.

Although you don't state it explicitly, the success of your method depends on molecules with similar property values clustering together. I think that you should verify this more explicitly, although I do appreciate that in Figure 2 you validated that structurally similar molecules cluster together, which is an important start to producing similar molecules given a target molecule.

One nitpick about Figure 2: you don't mention how these similarity values are calculated.

### Latent directions: regression becomes classification??

In section 5.2-5.3 your motivation seems to significantly switch from regression based (manipulating real-valued properties) to classification based (finding a hyperplane between positive and negative classes).
I find this awkward and not well explained **(major-)**.
In general, the existence of a separating hyperplane can only be guaranteed when the positive and negative examples form disjoint convex sets.
This seems unlikely to occur with generative models with a Gaussian prior.
It seems plausible to me that a generative model could have multiple latent directions which cause the property values to increase.
For example, consider molecular weight, which is modified by adding/removing/substituting atoms (effectively anything). Since there are many ways to add/remove atoms from a molecule, I imagine that such a property could be modified by several directions in latent space, and that there would not be a $(n-1)$ dimensional hyperplane separating "positive" and "negative" classes.

Even if there is just 1 direction of increase which would make the separating hyperplane assumption sensible, I find the separation into positive and negative classes quite artificial **(minor-)**. I could not find an explanation of this in your paper, but thankfully was able to find it in the code (thanks for making your code accessible and quite readable **(minor+)**). It looks like you choose the 2nd percentile of your dataset as the cutoff between positive and negative (so you will have very unbalanced classes). I am a unsure of why this choice was made: was it driven by performance or just an arbitrary choice? An explanation of these choices would make the paper stronger.

## Experiments

I was very happy to see the authors evaluate their method on 3 dataset and over 100 properties **(major+)**. Often I think papers of this sort test too few properties / datasets and I am glad that the authors did not do this. I think that the choice of experiments was generally good, but I have concerns about some choices that the authors made.

### Datasets

I was very happy with the authors' choice of datasets **(minor+)**.

### Baselines

I think the authors were too narrow in their choice of baselines **(major-)**: they compared their method against 2 simple methods which are identical in every way except for the choice of hyperplane. There are many other methods which I think would make competitive and sensible baselines, notably conditional VAEs and graph to graph translation methods like [8]. I acknowledge that these methods don't work with pretrained models, but as a reader I would want to know whether the SR/SSR scores are competitive in an absolute sense, or whether they are just competitive relative to other methods which use pre-trained models. _Having at least 1 competitive baseline would make the paper much stronger.

### Molecular properties

Although the authors chose a large number of properties to test, I felt that these properties were all superficial molecular descriptors from `rdkit.Descriptors.descList` which are intrinsically not as difficult to manipulate as real-world experimental properties such as drug activity **(major-)**. The properties mostly fall into just a few categories which can be considered separately.

- MolWt, logP, HeavyAtomCount, and similar properties are essentially a weighted sum over the atoms in a molecule. They can be trivially manipulated by adding and removing specific atoms from anywhere in the molecule.
- all properties after "NHOHCount" are fragment properties which count instances of particular substructures. They can be trivially manipulated by simply adding or removing instances of this substructure.
- QED is more complex, but is essentially a measure of whether a molecule fulfills a specific set of conditions for MolWt, logP, and counts of certain fragments. It is somewhat more complex to manipulate but still not too difficult in my opinion.
- FpDensityMorgan* are essentially a normalized count of the number of unique atomic environments in a molecule, and can be increased/decreased by adding/removing unique substructures (admittedly I'm <90% sure about this one)
- EState, Chi*, Kappa*, BalabanJ are some sort of graph theoretic properties which I don't have a good intuition for, so I don't really know how difficult they are to manipulate.

Because of the simplicity of these properties, I did not find the manipulations shown to be impressive. I think it would make the paper much stronger to consider more complex properties. My first suggestion is the goal-directed objectives from Guacamol [15] (which are mostly based on fingerprints and therefore still not super realistic). Preferably I would like to see results on at least 1 objective which is not "similarity" or "rediscovery" since those are too easy. My second suggestion would be molecular docking scores, since these are more realistic. You could consider using the "dockstring" package [16] for this. It would not surprise me if MolSpacE performed poorly on these more complex properties.

### Results

In general I liked the presentation of the results **(minor+)**, but have a few questions **(minor-)**.

1. In Table 1, the averages for SR are often a lot higher than all the 7 property values you show in the table (e.g. ~60 for ChEMBL HierVAE-M while all the other values are < 30). This implies that there are either outliers, or that the properties shown in the table are somehow atypical. Can you provide clarification? One suggestion might be to not show the average, but instead show the average rank of all the baselines. The average can be highly influenced by some tasks being easier than others.
2. Figure 3 is very unclear. Are the plots for a single property, or for the 7 properties you study? I am not sure what the figure is showing or how to interpret it.
3. In section 6.4, did you achieve similar results with the fragment properties, or any more complex properties?

## References

[1] Grammar Variational Autoencoder http://arxiv.org/abs/1703.01925

[2] Molecular de-novo design through deep reinforcement learning https://doi.org/10.1186/s13321-017-0235-x

[3] Generating Focused Molecule Libraries for Drug Discovery with Recurrent Neural Networks 10.1021/acscentsci.7b00512

[4] Optimization of Molecules via Deep Reinforcement Learning 10.1038/s41598-019-47148-x

[5] Sample-Efficient Optimization in the Latent Space of Deep Generative Models via Weighted Retraining https://proceedings.neurips.cc//paper/2020/hash/81e3225c6ad49623167a4309eb4b2e75-Abstract.html

[6] Constrained Bayesian optimization for automatic chemical design using variational autoencoders https://pubs.rsc.org/en/content/articlelanding/2020/sc/c9sc04026a

[7] Optimizing Molecules using Efficient Queries from Property Evaluations http://arxiv.org/abs/2011.01921

[8] Learning Multimodal Graph-to-Graph Translation for Molecular Optimization http://arxiv.org/abs/1812.01070

[9] Conditional Molecular Design with Deep Generative Models https://doi.org/10.1021/acs.jcim.8b00263

[10] GAN Dissection: Visualizing and Understanding Generative Adversarial Networks http://arxiv.org/abs/1811.10597

[11] Interpreting the Latent Space of GANs for Semantic Face Editing https://openaccess.thecvf.com/content_CVPR_2020/html/Shen_Interpreting_the_Latent_Space_of_GANs_for_Semantic_Face_Editing_CVPR_2020_paper.html

[12] On the "steerability" of generative adversarial networks http://arxiv.org/abs/1907.07171

[13] Controlling generative models with continuous factors of variations http://arxiv.org/abs/2001.10238

[14] https://www.ncbi.nlm.nih.gov/pmc/articles/PMC3869489/

[15] GuacaMol: Benchmarking Models for de Novo Molecular Design 10.1021/acs.jcim.8b00839

[16] https://arxiv.org/abs/2110.15486

**Summary Of The Paper:**

This paper proposes Molecular Space Explorer (MolSpacE), a method to generate molecules with continuously varying properties using a pre-trained latent variable generative model. It essentially involves 3 steps (although these steps were not clearly spelled out in the paper):

1. Sample many points from the model and evaluate their properties
2. For each property, train a SVM to predict the property given the the latent vector
3. Use the normal to this hyperplane as a "property manipulation" direction in latent space, which can be added a latent vector to change the property value of the decoded molecule.

**Summary Of The Review:**

Overall I am happy to see a paper addressing this worthwhile problem, but feel that there are too many issues to recommend acceptance, and therefore I must recommend rejection. In order to change my mind, I think the following would need to be addressed:

1. Clarify differences between your method and similar methods for discovering directions in latent space proposed in [10-13]
2. Further clarification or justification of finding a hyperplane using a classification method
3. Comparing against a set of non-trivial baselines and having superior, or at least reasonably similar performance

I also think the paper could be made much stronger by:

1. Using a more difficult set of molecular properties
2. Improving the definition of molecular manipulation and the metrics proposed

---

### Official Review · Reviewer_5bTH · 2021-11-03

**Correctness:** 2
**Technical Novelty And Significance:** 2
**Empirical Novelty And Significance:** 2
**Recommendation:** 3
**Confidence:** 4

**Main Review:**

### ===== Detailed pros: =====



1. Interpreting and dissecting generative models of molecules is an interesting and important direction, and one that so far has not been sufficiently explored. Moreover, the authors have built an interactive system which makes their work much more accessible.



2. The paper is mostly clear, and also includes helpful figures (Figure 1 gives an especially good overview; Figure 2 is useful to get some qualitative intuition).


### ===== Detailed cons: =====



1. The work makes several very strong assumptions about what a good latent space-based generative models is, without sufficient evidence or ablations that this makes sense. Overall, it's not clear what the takeaways should be, or how we should interpret the comparison between the models.

(a) Section 5 gives qualitative results showing that nearby latent points decode to similar molecules, and then goes on to assume that properties are linear functions of the latent space, which is a very strong assumption. It's not clear if here "property is linearly separable" is a good approximation of "property can be reliably predicted with a non-linear model" or "property is easy to control using optimization". One way to tackle this would be to show ablations, for example proving that if a property is hard to predict with a linear model, then it's also hard to predict it generally (given only the latent code), or that it's hard to optimize for it.

(b) The steerability metric assumes monotonic changes to the property, which is rather strict, and it's unclear if results would look different if one used a soft monotonicity requirement.

(c) To add to this, it's not clear if the streerability or separability metrics correlate with something useful down-stream, e.g. with optimization performance, or with the ability of the optimization algorithm to exploit (overfit) to property predictors during optimization [7].

(d) While the authors propose to do optimization by following the normal vector of the separation plane, that does not sound like a very powerful or practical optimization method (it is not compared to more general methods [1-6], and it may not be worthwhile to do this comparison, as I wouldn't expect this to do very well). One argument the authors make is efficiency, which is fair, although it's important to keep in mind that black-box optimization methods are sometimes also very efficient, e.g. when the property predictor is learned directly from the latent space, or it's learned from the molecule space with a very simple model (e.g. random forest on molecular fingerprints). Thus, I think it's hard for the optimization method proposed in this work to really compete with more general ones, and the paper should focus more on explanability/interpretability/predicting which models will be good to optimize in without running optimization.

(e) Finally, it's not clear what getting a bad result on the metrics proposed by the authors means. For example, the authors mention that CGVAE has a large gap between SSR and SR, meaning that if often produces molecules that are different but have the same property value. It's unclear to me if that's a bad thing.



2. Some statements about prior work are just not true.

(a) "(…) current methods are confined to a limited number of molecular properties, which hinders real-world applications in drug discovery and material science. For example, existing work only discover two molecular properties, penalized logP (…) and QED" - false, as many existing methods can optimize towards an arbitrary (in-vivo-computable) objective function [1-5]. For most of these models, I know that they are used in real-life drug-discovery projects in the pharma industry. While many other works do work on toy properties such as ones mentioned by the authors, it's important to note this is not always the case. Even objectives in the Guacamol benchmark [6] are designed to be more realistic that optimizing towards logP/QED by combining several properties (and occasionally other constraints). I think a more realistic discussion is needed here, referring to [1-6] and other papers as well. I would also suggest to pivot this work more strongly on the interpretability aspect (and its quantification), instead of claiming that it actually improves molecular optimization in itself.

(b) To add to the above, even the abstract says: "it is difficult to (…) customize the output molecule with desired properties", again suggesting that molecular optimization is hard in itself. As mentioned above, molecular optimization is well-studied, and just getting high values of the optimization objective is typically not hard, and often can be achieved with even relatively simple genetic algorithms. On the other hand, I agree with low interpretability of such techniques, and that they are not very well-suited for interactive design, so there is certainly room for methods similar to the one presented in this work if framed correctly.





### =====  Other comments: =====



- The paper keeps referring to "drug-like" and "drug-unlike" molecules, while actually talking about "high QED" and "low QED". It would be useful to make this explicit (drug-likeness is a vague and hard to define notion, while QED is a simple handcrafted approximation of that notion).

- Bottom of page 5 is confusing, as it shuffles may very strong assumptions without calling them out explicitly. First, it talks about manipulating multiple properties, but equations suggest it's talking about something more specific, which is manipulating *a linear combination* of several properties. It then states that property values follow a multivariate normal distribution, which is a very unexpected conclusion, and is only true because of the strong linearity assumption.

- Figure 4 is very hard to read, maybe use a differrent form to present this result.

- Section 6.3 mentions that MoFlow performed better than CGVAE on the metric defined by the authors due to its reversibility. I think given the vast amount of differences between the models, attributing the difference to reversibility is just speculation, so should be marked as such. Overall, that whole paragraph (2) is highly speculative and in my opinion doesn't really bring much.

- Table 1 is hard to read, because it compares both different models and different ways of choosing the latent direction at the same time. Maybe it would be more readable to first compare the ways of choosing latent directions, establish that the proposed way works the best, and then compare the models.

- It is very odd that the largest manipulation direction performs worse than random. It would be useful to get a bit more intuition into why that happens.

- Bottom of page 8 defines several ranges; what are these ranges for? Distance in the latent space?



### =====  Nits (didn't influence my score, just here to help): =====



- abstract: "generative models can synthesize new molecules" -> I guess you mean "propose" or "design", as "synthesize" would imply the molecules are actually made

- page 2: "the steering the" -> I guess replace the second "the" with "of"

- page 3: "latent space, which is usually modelled as a Gaussian distribution"; "latent space, which is commonly assumed to be Gaussian distribution" -> technically it's the prior that is gaussian, while the latent space is R^l (which is not a distribution)

- page 3: "and capable" -> "and is capable"

- page 3: "there exists property functions fP which defines" -> I'd drop the "s" in both cases

- page 5: "scaling the changes" -> "scales the changes"

- Equation 10: I'm assuming this \mathcal{E} is supposed to mean expectation; if so, then it's more common to use \mathbf{E} for that

- Many places use \mathcal{R} to denote the set of real numbers, while again it's common to use \mathbf{R}

- In Equation 8 the scaling factor alpha seems to turn into k in Equation 13 (and then alpha means something different)

- Page 8: "obverse" -> "observe"



### ===== References =====



[1] Efficient multi-objective molecular optimization in a continuous latent space

[2] Learning to Extend Molecular Scaffolds with Structural Motifs

[3] REINVENT 2.0 – an AI Tool for De Novo Drug Design

[4] Reinforced Molecular Optimization with Neighborhood-Controlled Grammars

[5] Hit and Lead Discovery with Explorative RL and Fragment-based Molecule Generation

[6] GuacaMol: Benchmarking Models for de Novo Molecular Design

[7] On failure modes in molecule generation and optimization

**Summary Of The Paper:**

This paper proposes to analyse latent space-based generative models of molecules by measuring how separable the latent space is with respect to several molecular properties. If a property is sufficiently separable, then it can be affected by moving along the normal vector of the separation plane. Experiments compare the streerability of several established generative models.

**Summary Of The Review:**

While the direction pursued by the paper is interesting, there are many strong assumptions made with little justification, and overall it's not clear how one should interpret the results in this work, or what the impact on realistic optimization tasks is. Finally, some things are also unclear, and the relation to prior work is somewhat misrepresented. Thus, I believe this paper falls below the acceptance threshold.

---

### Official Review · Reviewer_MABC · 2021-11-08

**Correctness:** 2
**Technical Novelty And Significance:** 2
**Empirical Novelty And Significance:** 1
**Recommendation:** 3
**Confidence:** 4

**Main Review:**

The paper uses linear models trained on latent dimensions of an existing deep generative model for molecular property prediction. The normal directions of the learned decision boundaries are then used to manipulate molecules.

Strengths:
(1) The paper compares proposed approach with random manipulation and Largest range manipulation on three different datasets, there different deep generative models, and different molecular properties.

Weaknesses:
(1) The idea of using linear models trained on latent dimensions of existing deep generative models for steerable molecule generation or for interpreting/interacting with generative models are not new. For example, see Das, et al. Nature Biomedical Engineering volume 5, pages 613–623 (2021).
(2) Steerable molecule generation using predictors trained on latent dimensions has been investigated before, e.g. Chenthamarakshan, et a. NeurIPS 2020, which has been shown to tackle multi-property manipulation well as well.
(3) The paper does not provide a comprehensive review of the existing works on steerable molecule generation, nor does it consider existing optimization/sampling based methods as baselines (Chenthamarakshan, et al, NeurIPS 2020; Fu, et al, AAAI 2018; Yang, et al, ICML 2020; arXiv:2011.01921).
(4) Current works consider many different complex molecular properties that are relevant for real-world applications, e.g. . Therefore, this statement does not hold "For example, existing work only discover two molecular properties, penalized logP (octanol-water partition coefficient) and QED (Drug-likeness) (Jin et al., 2018; Shi et al., 2020; Liu et al., 2018). " For example, https://arxiv.org/abs/1912.05910 considers DRD2 activity prediction.
(5) It is not clear what extent of structural diversity is considered in SSR estimation.

**Summary Of The Paper:**

This work investigates molecule manipulation towards desirable properties by leveraging existing deep generative models.

**Summary Of The Review:**

The idea of leveraging linear property predictors trained on latent dimensions of deep generative models for molecule manipulation/controlled molecule generation is not novel. The work does not consider comparison with. existing baselines, nor does it report performance on real-world property manipulation tasks (such as protein activity manipulation). It is not clear what is the real-world significance of continuous property change.

---

### Decision · Program_Chairs · 2022-01-20

**Decision:**

Reject

**Comment:**

The reviewers find the work to address an interesting and important problem but have several critical concerns about its insufficient treatment of prior work in this area,  lack of novelty in relation to the body of existing literature.